# Sarcomeric remodelling in human heart failure unraveled by single molecule long read sequencing

Jan Haas[1,2,3,13], Sarah Schudy[1,2,3,13], Benedikt Rauscher[4,13], Ana Muñoz Verdú[1,2,3], Steffen Roßkopf[1,2], Christoph Reich [1,2,3], Gizem Donmez Yalcin[1,2,3], Abdullah Yalcin [1,2,3], Timon Seeger[3,5], Christoph Dieterich [3,5], Manuel H Taft [6], Marc Freichel [3,7], Dirk Grimm [8,9], Dietmar Manstein[10], Johannes Backs [3], Norbert Frey[3,5], Lars Steinmetz[4,11,12] & Benjamin Meder [1,2,3]✉

## Abstract

Dysregulation of alternative splicing – mediated by factors such as RBM20 or SLM2 – can affect proper gene isoform control, disrupting gene isoform homeostasis and underpins severe cardiomyopathy in both animal models and patients. Although innovative therapies target various sarcomeric components, the impact of isoform switching in cardiac disease remains poorly understood. Here, we applied nanopore long-read sequencing to map the full-length transcriptome of left ventricular tissue from thirteen non-failing controls, ten patients with dilated cardiomyopathy (DCM), and ten with ischemic cardiomyopathy (ICM). Our analysis identified 78,520 transcripts, 31% of which represent novel isoforms of known genes. Notably, the transcriptomes of DCM and ICM were largely indistinguishable, indicating that end-stage heart failure is characterized by a convergent isoform landscape, irrespective of disease etiology. Among 11 prototypical sarcomere genes, 10 displayed highly significant isoform shifts ($p = 5.23 \times 10^{-45}$–$2.89 \times 10^{-200}$). Focusing on tropomyosin, we observed that while the predominant cardiac gene *TPM1* showed moderate up-regulation of its transcript isoforms, transcripts derived from *TPM3*—typically expressed at lower levels in the healthy heart—were markedly increased in heart failure.

**Keywords** Heart-Failure; Long-Read; Transcript Isoforms; Sarcomere; Tropomyosin
**Subject Categories** Cardiovascular System; Chromatin, Transcription & Genomics

## Introduction

Heart failure is a severe medical condition in which the heart is weakened and unable to pump blood properly (Ponikowski et al, 2016). It is caused by a variety of diseases: e.g., arrhythmia, toxic damage, ischemic heart disease, and cardiomyopathies. Cardiomyopathy is defined as heart muscle disease with structural and/or electrical dysfunction with four main subgroups, with dilated cardiomyopathy (DCM) being the most prevalent one (Maron et al, 2006; Elliott, 2023). Besides secondary causes, 30–50% of the DCM cases are known to be genetic (Hershberger et al, 2013) involving mostly sarcomeric and z-disc associated genes (McNally and Mestroni, 2017; Millat et al, 2011; van Spaendonck-Zwarts et al, 2013; Hershberger et al, 2018). Ischemic cardiomyopathy (ICM) derives from myocardial infarction or coronary artery disease (Dang et al, 2020). Underlying causes also involve environmental factors and genetics.

Several cases are known in which aberrant alternative splicing contributes to cardiomyopathies (Kong et al, 2010; Sweet et al, 2018). Splicing factors such as RNA-binding motif protein 20 (RBM20) determine the physiological mRNA landscape formation, and rare variants in the *RBM20* gene explain up to 6% of genetic DCM cases (Koelemen et al, 2021). Putative splice regulators with relevance during heart failure include Sam68-like mammalian protein 2 (SLM2) or RBFox1. In the human heart, SLM2 binds to important transcripts of sarcomere constituents, such as those encoding myosin light chain 2 (MYL2), troponin I3 (TNNI3), troponin T2 (TNNT2), tropomyosin 1/2 (TPM1/2), and titin (TTN) (Boeckel et al, 2022). Due to the complex structure and variety in the length of sarcomeric genes, the complete coverage of the exon junctions is challenging (Uapinyoying et al, 2020), leading to increased efforts to develop novel techniques to identify splicing events in a more accurate manner. Nanopore long-read sequencing is a rather new technique known as third-generation sequencing,

[1]Precision Digital Health Unit of the Department of Internal Medicine III, Heidelberg University, Heidelberg, Germany. [2]Informatics for Life, Heidelberg, Germany. [3]German Center for Cardiovascular Research (DZHK), partner site Heidelberg, Heidelberg, Germany. [4]Genome Biology Unit, European Molecular Biology Laboratory (EMBL), Heidelberg, Germany. [5]Department of Internal Medicine III, Heidelberg University, Heidelberg, Germany. [6]Institute for Biophysical Chemistry, Hannover Medical School, Fritz–Hartmann–Centre for Medical Research, 30625 Hannover, Germany. [7]Department of Pharmacology, University of Heidelberg, Heidelberg, Germany. [8]German Center for Infection Research (DZIF) and German Center for Cardiovascular Research (DZHK), partner site Heidelberg, Heidelberg, Germany. [9]Department of Infectious Diseases/Virology, Section Viral Vector Technologies, Medical Faculty, BioQuant, Heidelberg University, Heidelberg, Germany. [10]Division for Structural Biochemistry, Hannover Medical School, 30625 Hannover, Germany. [11]Department of Genetics, Stanford University School of Medicine, Stanford, CA, USA. [12]Stanford Genome Technology Center, Palo Alto, CA, USA. [13]These authors contributed equally: Jan Haas, Sarah Schudy, Benedikt Rauscher.✉E-mail: Benjamin.meder@med.uni-heidelberg.de

enabling researchers to sequence extremely long stretches of nucleic acids at single-nucleotide resolution (Wu et al, 2023).

The technology leverages nanoscale protein pores embedded in an electrically resistant polymer membrane, which act as biosensors to permit the passage of a single nucleic acid strand at a time. This enables the sequencing of extended read lengths, facilitating the differentiation of highly similar isoforms, such as tropomyosins.

Tropomyosins are actin-binding proteins in the sarcomere and are encoded by over 40 different isoforms (Li et al, 2011). In muscle cells, tropomyosin contributes to the regulation of sarcomere contraction by sliding along the actin filament in response to calcium ($Ca^{2+}$) binding to the troponin complex (composed of troponin C, T, I encoded by *TNNC1*, *TNNT2*, and *TNNI3*, respectively). This movement exposes the myosin-binding sites on the actin filament, enabling muscle contraction (Craig and Lehman, 2001). The functional isoforms of tropomyosins in the heart are not well defined and differ between studies and authors. Several reports have highlighted the varying contributions of *TPM1* gene products to heart disease, with multiple variants identified as causative factors (Walsh et al, 2017). During rat heart development, full-length isoforms of *TPM1* were identified using nanopore long-read sequencing, with specific exon expression patterns associated with distinct developmental stages (Cao et al, 2021). Functionally, it is known that tropomyosin isoforms impact the actin–myosin binding by altering the maximum speed of the myosin motors or their properties of calcium sensitivity (Nefedova et al, 2022; Farman et al, 2018; Pertici et al, 2021; Reindl et al, 2022).

In this study, we delineate for the first time the isoform landscape of different forms of heart failure by single-molecule sequencing. We show the remarkable changes in isoform expression, especially remodeling the sarcomere with its core components. Functionally, we show that different isoforms of tropomyosin have altered calcium sensitivity, reflecting their potential adaptation to the changes in cellular requirements.

## Results

### Gene expression analysis in heart failure patients

In order to unravel the transcriptomic landscape in heart failure patients, we have performed cDNA full-length sequencing for 13 controls, 10 DCM patients, and 10 ICM patients. Full-length transcriptome analysis was done using FLAMES (https://github.com/LuyiTian/FLAMES). During differential gene expression analysis (DESeq2), we established a stringent significance threshold through $p$ value permutation (100 iterations). When samples from heart failure patients (DCM and ICM) were compared to controls, out of 22,622 genes, 872 genes were significantly upregulated, while 2224 genes were significantly downregulated based on the calculated $p$ value cut-off of $3.2 \times 10^{-5}$ (Fig. 1A). Based on significance, the most downregulated genes in heart failure (HF) patients were *PDIA5* ($p$adj $= 6 \times 10^{-39}$), *ADAMTS4* ($p$adj $= 9$ $10^{-28}$), and *MMRN2* ($p$adj $= 6 \times 10^{-27}$). The most upregulated genes were

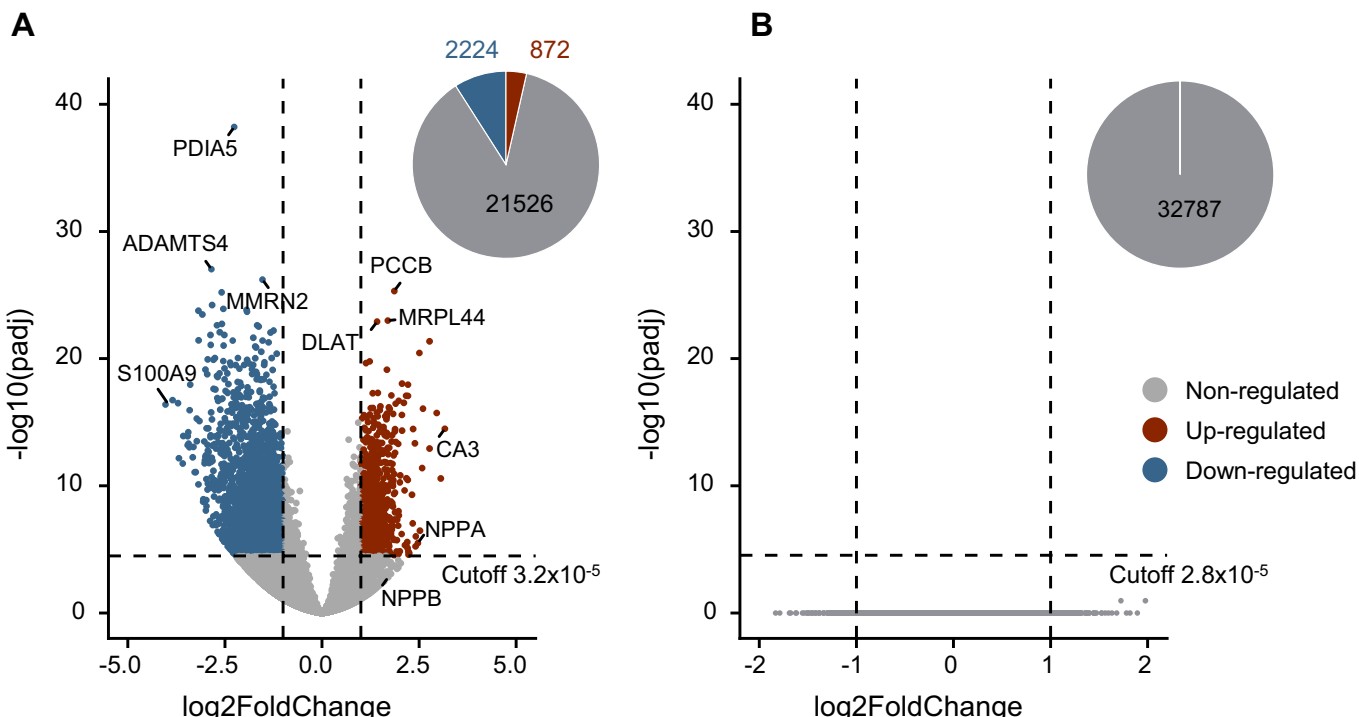

**Figure 1. Differentially expressed genes.**

(A) Volcano plots of heart failure (HF; $n = 20$) vs. control (CTRL; $n = 13$) and (B) dilated cardiomyopathy (DCM; $n = 10$) vs. ischemic cardiomyopathy (ICM; $n = 10$). Significantly upregulated, downregulated and non-regulated genes are depicted in red, blue, and gray, respectively. Vertical dashed lines mark the log fold change cutoff of $-1$ and $+1$ and the horizontally dashed line shows the significance cutoff, with $p$ values calculated by $p$ value permutation. The number of genes in the different groups are displayed in the pie chart. Statistical significance testing (Wald test) was done with DESeq2 to perform two-condition contrasts.

shown to be *PCCB* ($p$adj = $5 \times 10^{-26}$), *MRPL44* ($p$adj = $1 \times 10^{-23}$), and *DLAT* ($p$adj = $1 \times 10^{-23}$). The largest fold change was associated with an increase in gene expression of *CA3* (log2 foldChange = 3.16) and a decrease in gene expression of *S100A9* (log2 foldChange = −4.03) in HF patients (Fig. 1A). *NPPA*, a prototypical heart failure biomarker, was also strongly upregulated on its isoform level ($p$adj = $3 \times 10^{-6}$; log2 foldChange = 2.5). When comparing DCM and ICM, no differentially expressed genes were observed ($p$ value cutoff = $2.8 \times 10^{-5}$) (Fig. 1B).

## Long-read sequencing uncovers thousands of novel splicing events

In total, 78,520 isoforms were detected with long-read sequencing and, out of those, 31% ($n = 24{,}446$) were novel and 69% ($n = 54{,}074$) were known (Fig. 2A). The GENCODE transcriptome was used as a reference to determine the distribution of known and novel isoforms. The length is displayed on a log scale, revealing a prominent peak in the reference at ~2.8 kbp. The density of the known isoforms showed almost the same distribution, except for an additional larger peak around 3.4 kbp. However, novel isoforms tend to have the highest density for longer isoforms, which aligns with expectations based on the sequencing technology used (Fig. 2B). In our patient cohorts, the majority of genes had only one isoform annotated; however, genes with more than one isoform were frequently found (Fig. 2C). Most of the detected transcripts are multiexonic, indicating no bias towards smaller mono-exonic fragments (Fig. 2D). In addition, we determined the number of the most common alternative splicing events, including skipped exons (SE), alternative first (AF), and last (AL), as well as mutually exclusive (ME) exons and alternative 5' and 3' ends (A5 and A3, respectively). The splicing pattern was similar in novel compared to known isoforms, with the highest known counts at SE (novel = 3269; known = 11,300) and AF (novel = 2011; known = 9005). We also detected more A5 in known isoforms (novel = 2714; known = 4255); for A3 (novel = 1553; known = 4219); AL (novel = 1077; known = 2542); MX (novel = 542; known = 903); RI (novel = 825; known = 2402) (Fig. 2E).

Next, we used SQANTI3 (Tardaguila et al, 2018) to classify the long-read transcripts where in total 44,132 transcripts had a full-splice match (FSM), of which 78.5% are coding (Fig. 3A). Additionally, we investigated the length distribution (median length 1634 bp) which once again confirmed that there is no significant bias towards shorter fragments in the known isoforms with FSM compared to the novel ones in the catalog (NIC; median length 1862 bp). The latter contains new combinations of already annotated splice junctions or novel splice junctions formed from already annotated donors and acceptors. Furthermore, a third group was identified, designated "novel not in catalog" (NNC; median length 1341 bp), which are transcripts that use novel donors and/or acceptors (Fig. 3B). The median length for the transcripts with incomplete splice-match (ISM) was slightly shorter (1127 bp) due to the truncated reads. For the genic and fusion isoform groups (1193.5 bp and 1491 bp, respectively), the median length is similar to known and novel isoforms in the catalogue/not in the catalogue. The antisense (158 bp), genic intron (70 bp), and the intergenic region (107 bp) have very short median read lengths and have a higher possibility of noise (Fig. 3B).

## Differentially expressed transcripts in human heart tissue

During heart failure, a shift occurs in myosin heavy chain (MHC) isoform expression, characterized by a decrease in α-MyHC and an increase in β-MyHC in rodents; however, a previous study demonstrated that a considerable amount of α-MyHC mRNA is expressed in the normal human heart, and is decreased significantly in chronic end-stage heart failure patients (Nakao et al, 1997). Nevertheless, the detailed composition of the human sarcomere and the expression profiles of individual, mainly long isoform-encoding transcripts remain largely uncharacterized. Hence, we analyzed the expression and isoform landscape of important sarcomeric genes in control and heart failure individuals. Based on the long-read data, the significance cutoff was calculated by $p$-value permutation ($p$adj = $2.0 \times 10^{-5}$). Using this stringent cutoff, we identified 2468 upregulated and 4505 downregulated transcripts in DCM vs. CTRL (Fig. 4A) and 1749 upregulated and 3592 downregulated transcripts in ICM vs. CTRL ($p$ value = $2 \times 10^{-5}$) (Fig. 4B). Next, we performed a pathway analysis on upregulated isoforms in heart failure (DCM + ICM) vs. controls. As expected, cardiomyopathy pathways are significantly being enriched, e.g., "Dilated cardiomyopathy" (Enrichment FDR = 6.08e-05; fold enrichment = 3.98) or "Hypertrophic cardiomyopathy" (Enrichment FDR = 0.006; fold enrichment = 2.99), this is somehow expected, not surprising and yields little novel insight into the pathophysiology of heart failure. However, there are several pathways that are highly enriched and carry many genes in the pathways related to energy production metabolism (Fig. EV1; Dataset EV1). This finding is in line with our previous data on the dysregulation of several metabolites involved in glycolysis and the citric acid cycle (Haas et al, 2021). The importance of such pathways has also recently been reviewed, e.g., by Lopaschuk et al (Lopaschuk et al, 2021). Strikingly, no significantly differentially expressed transcript was detected between the two end-stage heart failure groups, again underlining that heart failure per se drives the transcriptional changes and not individual etiologies. Combining DCM and ICM as "heart failure (HF)", we find 2837 upregulated and 6305 downregulated isoforms compared to controls (Datasets EV2 and EV3). In Fig. 4C,D, two examples of the isoform analysis in sarcomeric genes are displayed. In the cardiac α-actin gene (*ACTC1*), a novel 5' end was detected and, five isoforms were differentially expressed in disease compared to control (Fig. 4C). For the myosin-binding protein C3 gene (*MYBPC3*), a novel coding isoform was identified, which is formed by intron retention and is significantly downregulated (Fig. 4D). As illustrated in the scheme in Fig. 4E, the majority of analyzed sarcomere transcripts exhibit significant changes. Of note, we found 37 coding genes with significant isoform changes in opposing directions, highlighting the plasticity in isoform-specific gene expression (Fig. EV2; Dataset EV4). For the "calcium voltage-gated channel auxiliary Subunit alpha2 delta 1" (*CACNA2D1*), which is involved in voltage-gated calcium channel activity and calcium channel regulator activity, we find four non-regulated (NO), four upregulated (UP) and one downregulated (DOWN) isoform. Figure EV3 gives an overview of the differential isoform usage of coding and non-coding isoforms (as determined by DRIMseq). Isoform "ENSG00000153956.15_82032797_82062969_1" (purple) is

**A**

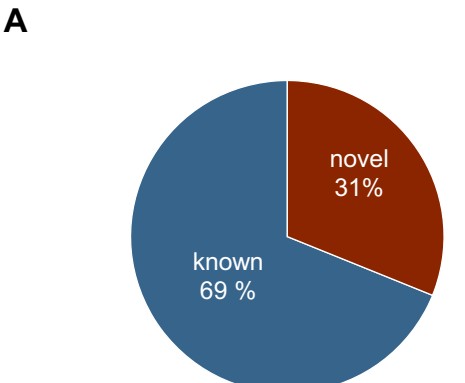

Total number of isoforms 78,520

**B**

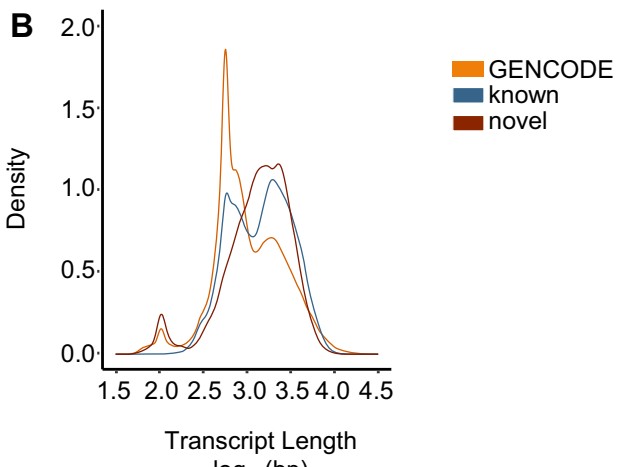

**C**

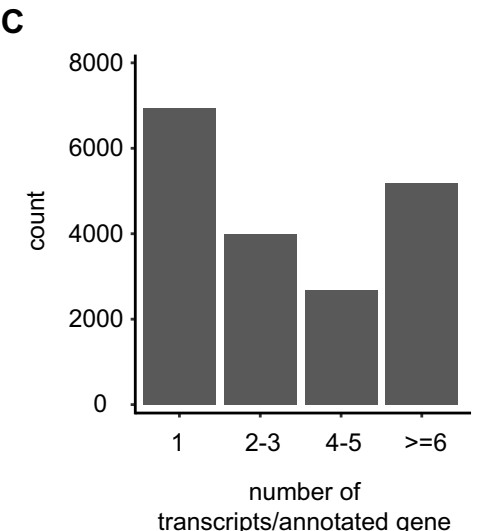

**D**

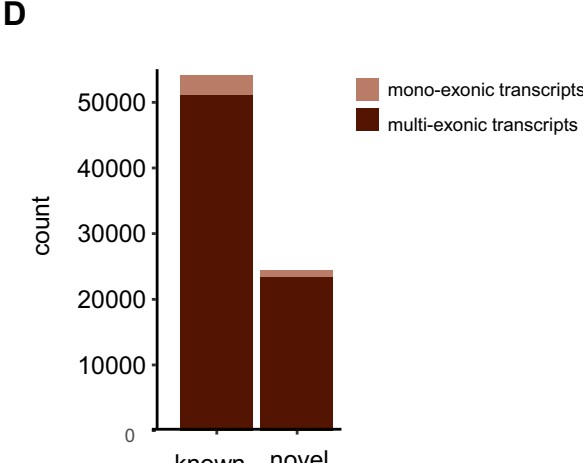

**E**

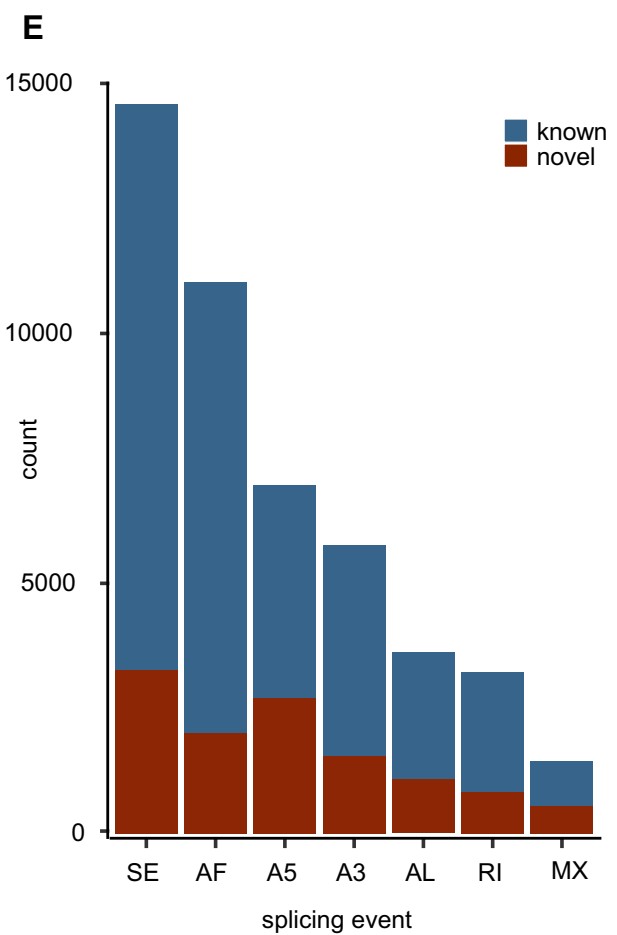

**Figure 2.   Long-read sequencing uncovers thousands of novel splicing events.**

(A) Total amount of transcripts identified by long-read sequencing, which detects 31% as novel transcripts. (B) Kernal density read-length distribution of reference GENCODE comprehensive gene annotation (.v33) transcripts (orange), known transcripts (blue), and novel transcripts (red). (C) Total counts of transcripts per annotated gene. (D) Total count of mono- and multiexonic transcript structure in known and novel transcripts. (E) Total count of splicing events as defined by SUPPA. SE skipped exon, AF alternative first exon, A5 alternative 5′ end, A3 alternative 3′ end, AL alternative last exon, RI retained intron, MX mutually exclusive exons.

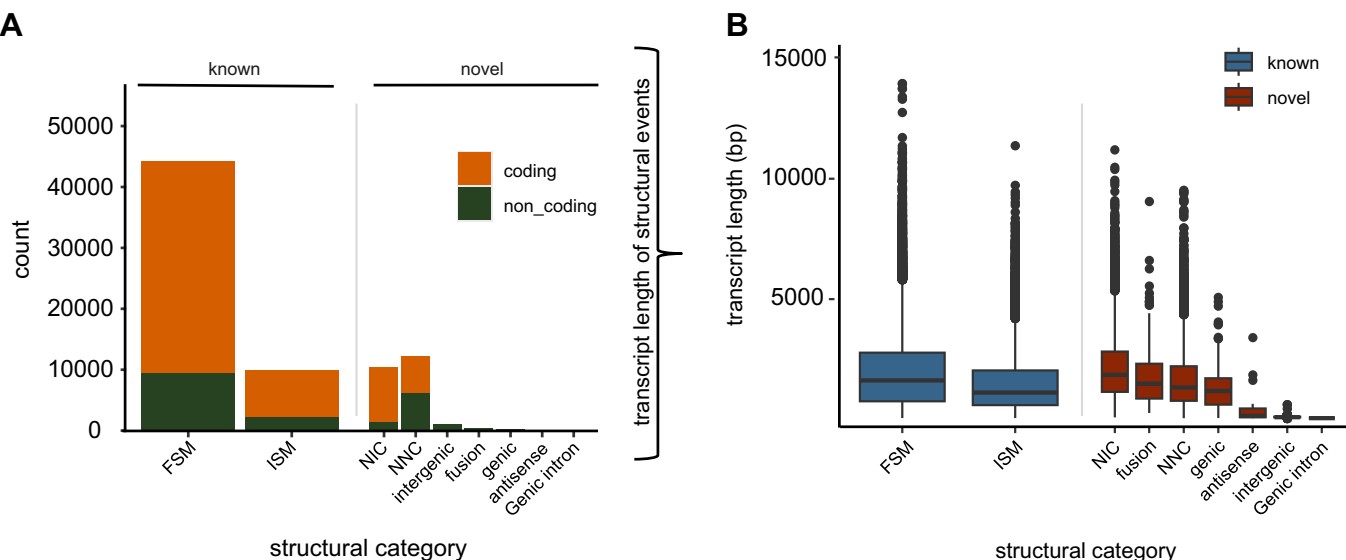

**Figure 3.   SQANTI classification of detected isoforms.**

(A) Total count (left) of structural events and (B) their length distribution (right) as defined by SQANTI. Known and novel isoforms are separated in coding (orange) and non-coding (green) isoforms. The known (blue) and novel (red) isoforms in the single structural categories were further analyzed by their length. Full-splice match: FSM, incomplete splice match: ISM, novel in catalog: NIC, novel not in catalog: NNC. Length of transcripts ranged as following for FSM ($n = 44,132$; min = 68; max = 13,956; median = 1634; q1 = 762; q3 = 2780; p95 = 6225); ISM ($n = 9942$; min = 64; max = 11,371; median = 1127; q1 = 604; q3 = 2044; p95 = 5226); NIC ($n = 10,393$; min = 94; max = 11,199; median = 1862; q1 = 1155; q3 = 2830; p95 = 5811); fusion ($n = 407$; min = 407; max = 9055; median = 1491; q1 = 878; q3 = 2320; p95 = 4855); NNC ($n = 12,238$; min = 63; max = 9525; median = 1341; q1 = 784; q3 = 2215; p95 = 4794); genic (min = 67; max = 5080; median = 1194; q1 = 623; q3 = 1717; p95 = 3806); antisense ($n = 24$; min = 58; max = 3407; median = 158; q1 = 103; q3 = 465; p95 = 2696); intergenic ($n = 1119$; min = 52; max = 628; median = 107; q1 = 99; q3 = 119; p95 = 186); genic intron ($n = 1$; min = 70; max = 70; median = 70; q1 = 70; q3 = 70; p95 = 70).

significantly downregulated in heart failure ($p$adj = 1.369823e-13), while isoforms "81946444_82410637_1" (orange), "81964331_82410637_1" (dark blue), "81965594_82410637_1" (pink), and "82032797_82410637_1" (light green) are upregulated.

## Tropomyosin 3 transcript usage is significantly different in heart failure vs. controls

Tropomyosins have been demonstrated to be integral to cardiomyocyte contractility and are of particular importance in mediating calcium-contraction coupling (Pieples et al, 2002). The four major tropomyosin genes—TPM1, TPM2, TPM3, and TPM4—produce a diverse array of isoforms through alternative splicing. The splicing of four key exons—1, 2, 6, and 9—varies across human tropomyosin isoforms and can be systematically represented using a four-letter code, where each letter corresponds to a specific splice variant of the respective exon. For example, "a.a.b.d" signifies that exon 1 is the splice form a, exon 2 is the splice form a, exon 6 is the splice form b, and exon 9 is the splice form d. If an exon is absent, it is represented by a dash (-), indicating a shorter isoform. For instance, "b.-.b.d" denotes the omission of exon 2, resulting in a

truncated Tpm variant. Historically, the isoforms TPM1, TPM2, and TPM3, primarily linked to muscle functions, were referred to as α-Tpm, β-Tpm, and γ-Tpm, respectively (Wieczorek, 2018), while TPM4 has been primarily linked to non-muscle functions. Notably, isoforms of TPM1 and TPM3 are also found in non-muscle cells, where they contribute to cytoskeletal organization and intracellular dynamics. While these genes have been extensively studied in animal models and during development, their full isoform diversity and specific roles in human tissues remain an area of ongoing research (Reindl et al, 2022; Schevzov et al, 2011). While mutations in the TPM3 are linked to congenital myopathies, the regulation of its major isoforms and their splice variants during heart disease remains poorly understood (Ohlsson et al, 2009). Our analysis revealed a significant prevalence of isoform switching in the tropomyosin genes in subjects diagnosed with heart failure, when juxtaposed with the control group. Under physiological conditions, TPM1 is expressed at higher levels than TPM3, and during HF, the TPM1-207 transcript encoding the long isoform Tpm1.1 (a.b.b.a) is specifically upregulated (Fig. 5A). Intriguingly, TPM3-224, which encodes the long isoform Tpm3.12 (a.b.b.a), displays much higher expression levels in HF, almost reaching levels of TPM1 (Fig. 5A),

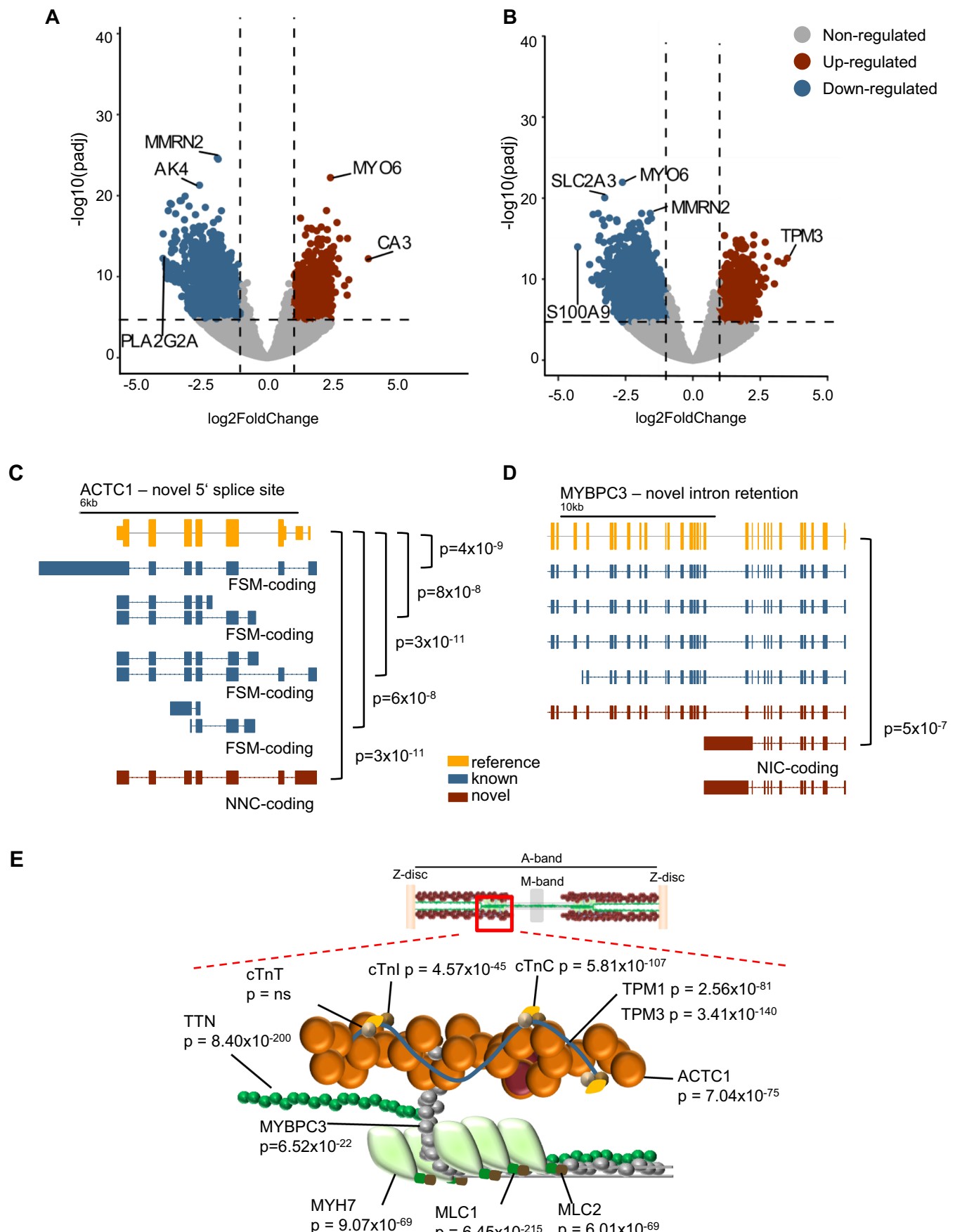

**Figure 4.  Differential transcript expression in heart failure.**

(A) Volcano plot of differentially expressed transcripts in DCM ($n = 10$) vs. CTRL ($n = 13$) and (B) volcano plots of differentially expressed transcripts in ICM ($n = 10$) vs. CTRL ($n = 13$). Significantly upregulated, downregulated, and non-regulated genes are depicted in red, blue and gray, respectively. Vertically dashed lines mark the log-fold change cutoff of $-1$ and $+1$, and the horizontally dashed line shows the significance cutoff, with $p$ values calculated by $p$ value permutation. Statistical significance testing (Wald test) was done with DESeq2 to perform two-condition contrasts. (C) Schematic transcript structure example of ACTC1. (D) Schematic transcript structure example of MYPC3. Reference gene, known isoform and novel isoform are shown in orange, blue, and red, respectively. The $p$ values indicate the significant differentially regulated isoforms of heart failure vs. CTRL. FSM, NNC, and NIC refer to SQANTI's structural analysis and is only shown for the differentially expressed transcripts. (E) Scheme displaying that the most studied sarcomere transcripts show significant changes in HF vs. CTRL. Sarcomere graphics were adapted from (Sedaghat-Hamedani et al, 2018).

resulting in an even higher fold change of gene expression in HF (Fig. 5B). A comparison of the transcripts *TPM3-224 (Tpm3.12)*, and *TPM1-207 (Tpm1.1)* reveals that they are paralogs with highly conserved intron-exon structures (Fig. 5C). Both genes are also highly conserved on the protein level, as shown by protein alignment (Appendix Figs. S1 and S2). An analysis of the variable amino acids revealed that Tpm3.12 (TPM3-224) lacks putative or proven serine/threonine phosphorylation residues at seven locations but has an additional three serine/threonine residues at other locations, which might circumvent posttranslational regulatory mechanisms.

We next analyzed the precise composition of the isoforms of the HF-related *TPM3* upregulation. In healthy controls, *TPM3-206* (encoding the short isoform Tpm3.2 (b.-.b.d) and *TPM3-212* (encoding the short isoform Tpm3.1 (b.-.a.d)) are the predominantly produced transcripts. However, in the disease state, a notable shift in isoform expression occurs, with increased production of *TPM3-210*, which encodes Tpm3.13 (a.b.a.d), and *TPM3-224*, which encodes Tpm3.12 (a.b.b.a) (Fig. 6A). In addition, transcript abundance in transcripts per million (TPM) was assessed for the four transcripts in CTRL, DCM, and ICM samples. In accordance with the differential gene expression analysis based on the negative binomial distribution (DESeq), we detected significant differences in HF vs. CTRL for *TPM3-224 (Tpm3.12)*: $\log_2$ fold change $= 3.3$, $p$ value $= 1.5 \times 10^{-18}$ and *TPM3-210 (Tpm3-isoform 13)*: log2 fold change $= 3.1$, $p$ value $= 1.8 \times 10^{-17}$. In DCM and ICM vs. CTRL, a significant upregulation was detected for *TPM3-224 (Tpm3.12)*: $\log_2$ fold change $= 3.03$, $p$ value $= 2.5 \times 10^{-12}$ (DCM) and *TPM3-224 (Tpm3.12)*: $\log_2$ fold change $= 3.5$, $p$ value $= 6 \times 10^{-16}$ (ICM) and *TPM3-210 (Tpm3-isoform 13)*: log2 fold change $= 2.9$, $p$ value $= 1.8 \times 10^{-11}$ (DCM) and *TPM3-210 (Tpm3-isoform 13)*: log2 fold change $= 3.4$, $p$ value $= 3.3 \times 10^{-15}$ (ICM) was shown (Fig. 6B). The differences between these transcripts arise from variations in exon 1, exon 6, and exon 9 (Fig. 6C). According to official protein nomenclature, Tpm3.2 and Tpm3.1, the products of TPM3-206 and TPM3-212, respectively, are both classified as cytoplasmic tropomyosin isoforms. In contrast, Tpm3.12, encoded by TPM3-224, has been detected in striated muscle, whereas Tpm3.13, the product of TPM3-210, was previously reported only in rat and mouse but is now newly identified in human tissue (Geeves et al, 2015).

To assess, whether increased TPM3 transcript levels result in elevated protein levels of this gene, we have performed a meta-analysis on recently published studies based on mass-spectrometry protein quantification (Chen et al, 2018; Hunter et al, 2024; Jani et al, 2025). Significantly higher protein levels in heart failure patients were found for TPM3 (Log2-FC $= 0.33$ [0.10,0.56]), which means a 26% higher TPM3 expression in HF, confirming our transcript-based findings (Fig. 6D). For TPM1, the analyzed studies showed quite heterogeneous protein expression (Log2-FC $= -0.08$ [$-0.54$,0.38]) (Fig. 6E).

To explore the transcriptome-wide and proteome-wide correlation, we conducted a nine-sector analysis, which reflects well-known mechanisms of mRNA-to-protein correlations. As shown in Fig. 6F, we find non-regulated (sector 5) as well as concordant (3, 7) and discordant (1, 9) candidates. In total, 527 of 4833 (11%) analyzed genes/proteins were concordantly regulated as a consequence of heart failure, including TPM3 (sector 3) (Fig. 6F).

## Tpm3.12 (Tpm3-224) shows enhanced calcium responsiveness compared to Tpm1.1 (Tpm1-207)

The regulatory effect of tropomyosins on actin–myosin interaction can be investigated using in vitro motility assays, which measure the parameters such as the sliding velocity of actin and actin–tropomyosin filaments, driven by myosin motors (Kopylova et al, 2013; Kron and Spudich, 1986). Therefore, we established and validated motility assays to examine the effects of Tpm1.1 and Tpm3.12 under varying calcium conditions (Fig. 7A). Using the motility assay, the $v_{max}$ for Tpm1.1 was calculated to be 780.1 nm/s, and the $pCa_{50}$ was 6.3. For Tpm3.12, $v_{max}$ was 728.5 nm/s, and the $pCa_{50}$ was 6.6 (Fig. 7B). The increased velocity for Tpm3.12 (TPM3-224) has been documented by video datasets for TPM3.12 at pCa 4 (Movie EV1), for TPM3.12 at pCa 7 (Movie EV2) and for TPM3.12 at pCa 9 (Movie EV3) in comparison to Tpm1.1 (TPM1-207) for TPM1.1 at pCa 4 (Movie EV4), for TPM1.1 at pCa 7 (Movie EV5) and for TPM1.1 at pCa 9 (Movie EV6). All in vitro–synthesized proteins were separated on SDS-PAGE according to their molecular weights (Fig. 7C).

## Discussion

Earlier investigations of the transcriptome reported distinct changes during heart failure (Sweet et al, 2018; Zhu et al, 2022). While these studies relied on short-read sequencing with its high base-call accuracy (Shumate et al, 2022), correct mapping of isoforms is not precisely possible due to read lengths below 300 bp and large sequence homology between isoforms. Nanopore long-read sequencing technology has emerged as a powerful and evolving technique for the analysis of such isoform-specific transcriptome, enlightening alternative splicing and abundance of novel transcripts.

The regulation of cardiac transcript isoforms is governed by a complex interplay of splicing factors, cofactors, and upstream signaling pathways, which are essential for maintaining cardiac function. Central to this process are key splicing regulators, such as RBM20, SLM2, hnRNP A1, and RBFox1, whose interactions shape

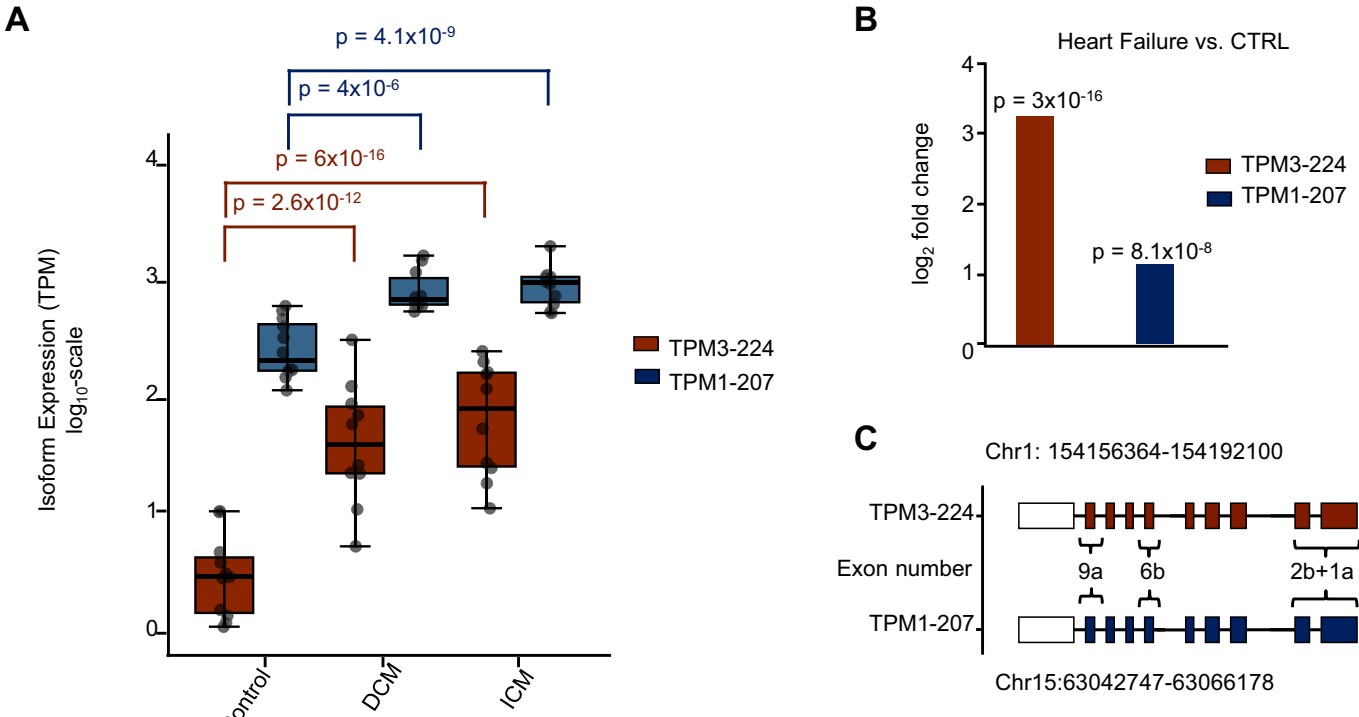

**Figure 5. TPM3-224 (Tpm3.12) is more upregulated in heart failure compared to TPM1-207 (Tpm1.1).**

(A) Isoform expression in transcript per million (TPM) of TPM3-224 (Tpm3.12) (red) and TPM1-207 (Tpm1.1) (blue) in CTRL ($n = 13$), DCM ($n = 10$), and ICM ($n = 10$). Significance is calculated by the Mann–Whitney U-test. p values are adjusted according to the Benjamini–Hochberg procedure, and significance was accepted as $p \leq 0.05$. Values ranged as following for TPM3-224-CTRL (min = 0; max = 1.04; median = 0.469; q1 = 0.0889; q3 = 0.601; p95 = 1.04); TPM3-224-DCM (min = 0.740; max = 2.50; median = 1.61; q1 = 1.36; q3 = 1.93; p95 = 2.43); TPM3-224-ICM (min = 1.06; max = 2.40; median = 1.91; q1 = 1.42; q3 = 2.22; p95 = 2.39); TPM1-207 (min = 1.71; max = 2.79; median = 2.25; q1 = 2.23; q3 = 2.62; p95 = 2.78); TPM1-207-DCM (min = 2.74; max = 3.22; median = 2.85; q1 = 2.80; q3 = 2.62; p95 = 2.78); TPM1-207-ICM (min = 2.73; max = 3.22; median = 2.85; q1 = 2.82; q3 = 3.30; p95 = 3.26). (B) Significant $\log_2$ fold change of TPM3-224 (Tpm3.12) (red) and TPM1-207 (Tpm1.1) (blue) in heart failure (DCM + ICM) compared to controls as determined by DESeq2. p values are Benjamini–Hochberg adjusted, and the p value cutoff = $2.3 \times 10$ was calculated by p value permutation. Statistical significance testing (Wald test) was done with DESeq2 to perform two-condition contrasts. (C) Highly similar isoform structure of TPM1-207 (Tpm1.1) and TPM3-224 (Tpm3.12) with nomenclature of important exons. Blue and red are in accordance with previously used colors for TPM1-207 (Tpm1.1) and TPM3-224 (Tpm3.12), respectively. The white square indicates the UTR.

the expression of critical sarcomeric proteins, particularly the tropomyosin isoforms (Table 1; Fig. EV4). RBM20, for instance, represses *TPM3* gene products by recruiting U2 snRNP and PTBP1, thus favoring the expression of the cardiac-specific *TPM1-207* (Tpm1.1) transcript. This regulation of tropomyosin isoforms is pivotal for maintaining proper sarcomere function in the heart. In contrast, SLM2 and hnRNP A1 promote the expression of non-muscle *TPM3* gene products by disrupting U2AF65 recruitment, highlighting their role in altering the balance between muscle and non-muscle isoforms in response to various physiological and pathological conditions. Furthermore, RBFox1 enhances *TPM1-207* (Tpm1.1) expression by stabilizing the U1 snRNP complex and binding to UGCAUG enhancer motifs in sarcomeric transcripts, further ensuring the maintenance of sarcomere integrity.

Upstream signaling pathways, including those mediated by PKC, ERK/MAPK, and calcium signaling, also play a critical role in regulating the splicing of TPM isoforms. RBM20 is known to be regulated by both PKC and ERK/MAPK signaling, ensuring that *TPM1-207* (Tpm1.1) remains the dominant isoform in the left ventricle of a normally functioning heart. The activation of SLM2 by ERK/MAPK and calcium signaling pathways leads to an increase

in TPM3 isoform expression, which is particularly important in the context of cardiac stress. Additionally, TGF-β signaling and the RhoA/ROCK pathway regulate TPM3 isoforms, especially during fibrosis and post-injury remodeling, underscoring the dynamic nature of isoform expression during heart disease.

The relevance of these splicing mechanisms extends beyond basic biology, with mutations and dysregulation of these factors contributing to cardiac disease. Mutations in RBM20, for example, lead to a shift from *TPM1-207 (Tpm1.1)* to *TPM3* transcripts, weakening sarcomere function and contributing to dilated cardiomyopathy (DCM) and heart failure. Similarly, overexpression of SLM2 in heart failure results in increased levels of *TPM3* gene products, driving cytoskeletal remodeling and fibrosis. In cases of cardiac hypertrophy, downregulation of RBFox1 results in the loss of *TPM1-207 (Tpm1.1)*, leading to compromised sarcomere integrity and activation of fibroblasts, further contributing to the progression of heart disease.

Recent advances in transcriptome analysis, particularly through the use of long-read sequencing technologies, have provided deeper insights into the complexity of cardiac splicing regulation. Previous studies on RBM20 mutant hiPS-CM, analyzed with FulQuant (Zhu

**A**

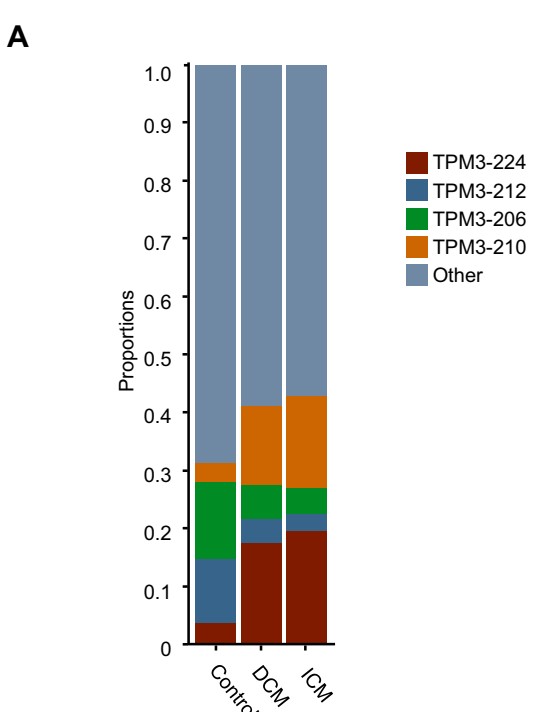

**B**

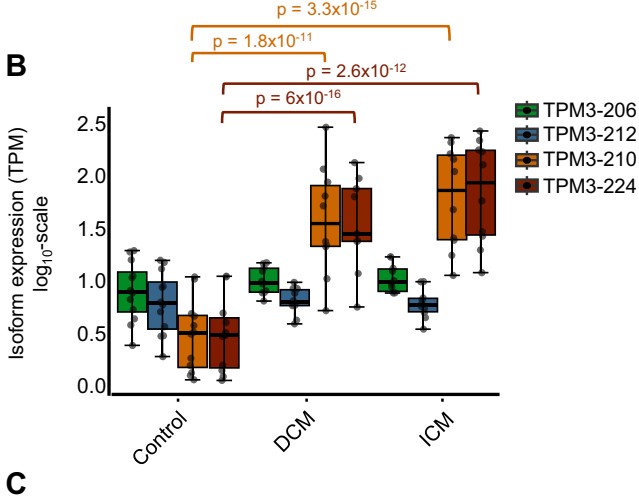

**C**

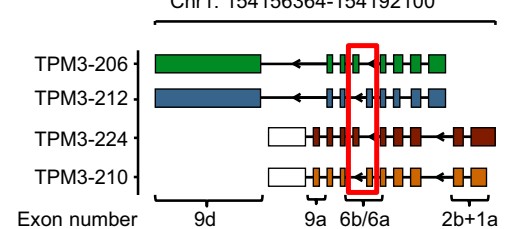

**D**

TPM3

| Study | | Estimate [95% CI] |
|---|---|---|
| Chen et al. | | 0.42 [ 0.03, 0.80] |
| Hunter et al. | | 0.27 [-0.02, 0.55] |
| Jani et al. | | 1.50 [-1.07, 4.07] |
| Random-Effects Model | | 0.33 [ 0.10, 0.56] |

Log₂ Fold Change

**E**

TPM1

| Study | | Estimate [95% CI] |
|---|---|---|
| Chen et al. | | 0.07 [-0.31, 0.46] |
| Hunter et al. | | -0.34 [-0.59, -0.09] |
| Jani et al. | | 2.26 [-0.31, 4.83] |
| Random-Effects Model | | -0.08 [-0.54, 0.38] |

Log₂ Fold Change

**F**

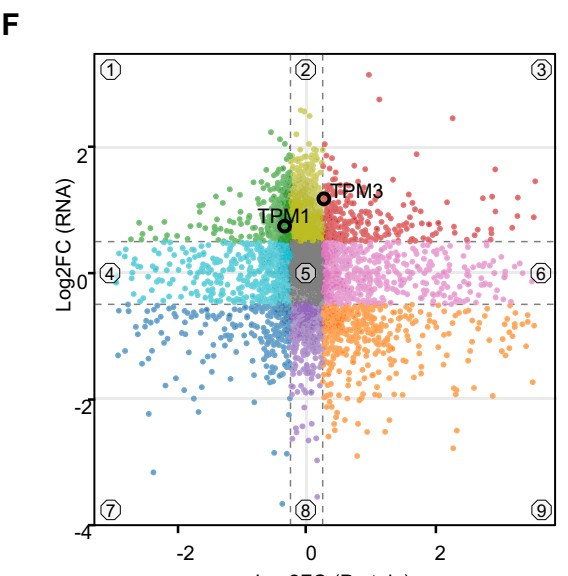

**Figure 6. TPM3-224 (Tpm3.12) is upregulated in heart failure.**

(A) Proportions of *TPM3* transcripts, calculated using DRIMSeq. (B) Isoform expression levels measured in transcript per million (TPM) for CTRL ($n = 13$), DCM ($n = 10$), and ICM ($n = 10$). Statistical significance was determined by using the Mann–Whitney *U*-test. Values ranged as following for TPM3-206-CTRL (min = 0.355; max = 1.26; median = 0.879; q1 = 0.696; q3 = 1.17; p95 = 1.26); TPM3-206-DCM (min = 0.778; max = 1.14; median = 0.953; q1 = 0.862; q3 = 1.09; p95 = 1.14); TPM3-206-ICM (min = 0.854; max = 1.20; median = 0.954; q1 = 0.870; q3 = 1.08; p95 = 1.20); TPM3-212-CTRL (min = 0.248; max = 1.06; median = 0.766; q1 = 0.533; q3 = 1.06; p95 = 1.16); TPM3-212-DCM (min = 0.560; max = 0.955; median = 0.762; q1 = 0.734; q3 = 0.885; p95 = 0.945); TPM3-212-ICM (min = 0.510; max = 0.962; median = 0.742; q1 = 0.675; q3 = 0.803; p95 = 0.962); TPM3-210-CTRL (min = 0.0513; max = 1.02; median = 0.484; q1 = 0.186; q3 = 0.654; p95 = 1.20); TPM3-206-DCM (min = 0.778; max = 1.14; median = 0.953; q1 = 0.862; q3 = 1.09; p95 = 1.14); TPM3-206-ICM (min = 0.854; max = 1.20; median = 0.954; q1 = 0.870; q3 = 1.08; p95 = 1.20); TPM3-224-CTRL (min = 0; max = 1.04; median = 0.469; q1 = 0.0889; q3 = 0.601; p95 = 1.04); TPM3-224-DCM (min = 0.740; max = 2.50; median = 1.61; q1 = 1.36; q3 = 1.93; p95 = 2.43); TPM3-224-ICM (min = 1.06; max = 2.40; median = 1.91; q1 = 1.42; q3 = 2.22; p95 = 2.39). (C) Isoform structures highlighting the nomenclature of key exons. Red squares indicate exon 6. For the meta-analysis, we used public data (Chen et al, 2018; Hunter et al, 2024; Jani et al, 2025), as described in the methods. (D) Forest plot showing $\log_2$ fold changes in TPM3 abundance (heart failure vs. normal) across three proteomics datasets. TPM3 is significantly upregulated in heart failure (pooled $\log_2$ fold change = 0.33, 95% CI: 0.10–0.56; $p = 0.004$; $I^2 = 0\%$). (E) Forest plot for TPM1 shows no consistent differential expression (random-effects model estimate = −0.08, 95% CI: −0.54 to 0.38; $p = 0.73$; $I^2 = 62.3\%$), indicating substantial heterogeneity across studies. Sample sizes (heart failure [HF+] vs. control [HF−]) were: Chen et al (23 vs. 11), Hunter et al (57 vs. 20), and Jani et al (10 vs. 10). Effect sizes and confidence intervals were derived from linear models adjusted for age and sex, followed by inverse-variance weighted random-effects meta-analysis. Positive Log2 fold change values indicate higher expression in heart failure. (F) Nine-sector scatter-plot association analysis. Log2 fold changes between heart failure and controls in RNA and protein are correlated.

et al, 2022), and the pig cardiac transcriptome, examined with StringTie (Müller et al, 2022), have identified thousands of isoforms previously unknown. In our own study, we analyzed the cardiac transcriptomes of more than 30 human individuals, providing the most detailed and unbiased view of the human isoform landscape in the heart. Our stringent cut-offs, designed to reduce the risk of false positives, combined with the use of the FLAMES pipeline (Fig. EV5) after careful benchmarking of available tools, ensured the reliability of our findings. These protocols not only enhance our understanding of the human cardiac isoform landscape but also pave the way for the application of nanopore sequencing in human diagnostic testing (Sedaghat-Hamedani et al, 2018).

As demonstrated in transcript redundancy analysis, *PCCB*, *MRPL44, DLAT, CA3*, and *NPPA* were found to be among the most significantly upregulated genes in HF. Propionyl-CoA carboxylase (PCC) is a dodecameric enzyme complex, consisting of 6α- and 6β-subunits, which are encoded by the *PCCA* and *PCCB* genes. The deficiency of PCC, which is encoded by *PCCB*, leads to propionic acidemia, resulting in DCM (Riemersma et al, 2017). The protein product of the *MRPL44* gene, ml44, was identified as a component of the large subunit of the mitochondrial ribosome (mitoribosome) and pathogenic variants in *MRPL44* causes infantile cardiomyopathy due to a mitochondrial translation defect (Friederich et al, 2021). Carbonic anhydrase (CA) is a zinc-containing enzyme that catalyzes the reversible hydration of carbon dioxide and plays a role in regulating the transport of bicarbonate (Lindskog, 1997). In a recent study, CA2 and CA3 were found to be more highly expressed in the HF group compared to controls, as determined by ELISA (Su et al, 2021). Carbonic Anhydrase 3 (CA3) was shown to be required for cardiac repair post-myocardial infarction via the Smad7-Smad2/3 signaling pathway (Su et al, 2024). In this study, the authors observed that CA3 deficiency restrains collagen synthesis, cell migration and gel contraction of cardiac fibroblasts, whereas overexpression of CA3 improves wound healing and cardiac fibroblast activation. Accordingly, the detected upregulation of CA3 in our DCM patients might point to an activation of a repair mechanism. Dihydrolipoamide *S*-acetyltransferase (*DLAT*) encodes the component E2 of the multi-enzyme pyruvate dehydrogenase complex (PDC). PDC resides in the inner mitochondrial membrane and catalyzes the conversion of pyruvate to acetyl coenzyme A. Interestingly, we have previously identified altered pyruvate

metabolite levels in an ambulatory HF cohort (Haas et al, 2021). Both *NPPA* and *NPPB* are induced by cardiac stress and serve as markers for cardiovascular dysfunction or injury (Giovou et al, 2024). In our analysis, the *NPPA* gene was found to be significantly upregulated while the *NPPB* gene remained unaltered.

Among the strongly downregulated genes, *ADAMTS4* and *S100A9* were both reported in previous studies as cardiac injury markers. The disintegrin-like and metalloproteinase with thrombospondin motif (ADAMTS) family comprises 19 proteases that regulate the structure and function of extracellular proteins in the extracellular matrix and blood. A number of studies have also investigated the potential role of ADAMTS-1, −4, and −5 in cardiovascular disease (Santamaria and de Groot, 2020). *ADAMTS4* was identified as a novel adult cardiac injury biomarker with therapeutic implications in patients with cardiac injuries. It is also downregulated, as observed in our study, and depicted among the hub genes identified in potential DCM-related targets by meta-analysis and co-expression analysis of human RNA-sequencing datasets (Yuan et al, 2022).

During HF, the isoform switch of myosin heavy chains (MHC) plays a crucial role in the alteration of cardiac contractility. In a healthy human heart, the predominant MHC isoform is beta-myosin heavy chain (β-MHC), which supports efficient contraction due to its rapid ATPase activity. The expression of α-MHC in the human heart is very low, whereas in rodents, the majorly expressed isoform is α-MHC. However, in rodents with heart failure, a shift occurs toward the expression of the beta-myosin heavy chain (β-MHC), a slower isoform associated with reduced contractile function (Liu et al, 2016; van der Velden et al, 2003). This isoform transition is believed to be a compensatory response to maintain cardiac function under stress, but it leads to a decline in the heart's efficiency and performance. The molecular mechanisms underlying this isoform switch involve changes in the transcriptional regulation of the MHC genes, including alterations in the expression of factors such as myocyte enhancer factor 2 (MEF2) and the calcium/calmodulin-dependent kinase II (CaMKII) pathway, both of which were implicated in promoting β-MHC expression in the failing heart (Sato et al, 2004). We observed in this study a profound change of sarcomeric isoforms that might functionally impact cardiomyocytes in a similar manner. Alternative splicing of *TPM* genes has been previously reported to generate tropomyosin isoforms with distinct biochemical and

**A**

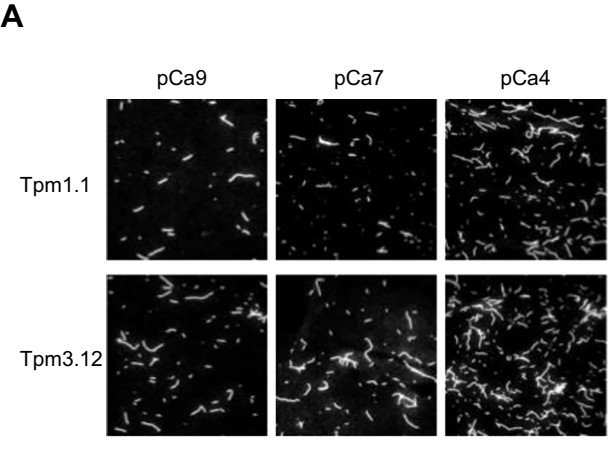

**B**

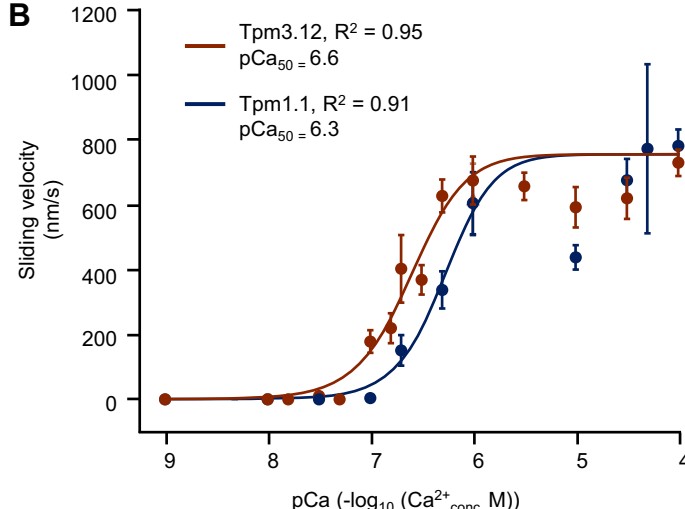

**C**

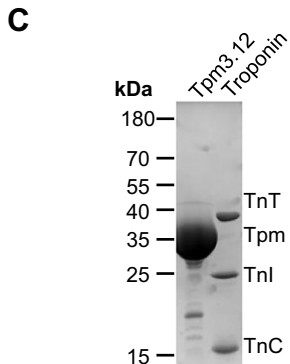

**Figure 7. In vitro motility assays were performed in the presence of Tpm1.1 (TPM1-207) and Tpm3.12 (TPM3-224).**

(A) Still image from the movies (Movies EV1–6) covering the motility assays at different calcium concentrations of the two investigated tropomyosin isoforms Tpm1.1 (TPM1-207) and Tpm3.12 (TPM3-224). Movies were captured with a 60x lens with an intermediate magnification of 1.5. The white bar in the upper left corner scales to 10 μm. (B) Calcium dose-response curve from in vitro motility assays for Tpm1.1 (TPM1-207) (blue) and Tpm3.12 (TPM3-224) (red). The coefficient of determination ($R^2$) is provided for the Hill function fit. pCa values are expressed as $-\log_{10}$ of calcium concentrations (Molar) with $pCa_{50}$ indicating the calcium concentration at half–maximal velocity. Sliding velocity values are presented as mean ± SEM. For the different calcium concentrations, the following number of tests have been performed for TPM3-224: pca9 = 12; pca8 = 20; pca78 = 4; pca75 = 12; pca73 = 5; pca7 = 47; pca68 = 16; pca67 = 14; pca65 = 33; pca63 = 22; pca6 = 37; pca55 = 32; pca53 = 6; pca5 = 35; pca45 = 28; pca4 = 71 and for TPM1-207 pca9 = 9;pca8 = 11;pca75 = 6;pca7 = 30;pca68 = 17;pca67 = 11;pca65 = 20;pca63 = 15;pca6 = 24;pca55 = 13;pca53 = 10; pca5 = 20;pca45 = 23;pca43 = 6;pca4 = 48. (C) Proteins used for in vitro motility assays. SDS-PAGE of purified and reconstituted proteins are depicted by Coomassie staining. Expected molecular weights: Tpm3.12 (TPM3-224) (32.9 kDa), Troponin T (TnT, 34.5 kDa), Troponin C (TnC, 18.4 kDa), Troponin I (TnI, 24 kDa), Tpm1.1 (TPM1-207) (32.7 kDa), heavy meromyosin (HMM, 165 kDa), essential light chain (ELC, 22 kDa), and regulatory light chain (RLC, 16 kDa). PageRuler™ (Thermo Fisher Scientific) was used as a molecular weight marker. Source data are available online for this figure.

functional properties (Moraczewska, 2020; Kengyel et al, 2024; Pathan-Chhatbar et al, 2018; Rajan et al, 2010). Previous investigations suggested only limited expression of *TPM3* in the heart (Dube et al, 2020; Tucholski et al, 2020; Lawlor et al, 2010). By performing single-nucleus RNA sequencing of nearly 600,000 nuclei at a single-cell resolution, *TPM3* expression was specifically detected in cardiomyocytes across samples from 11 DCM and 15 HCM patients; as well as 16 nonfailing hearts (Chaffin et al, 2022). This observation suggested that *TPM3* is expressed in cardiomyocytes across both diseased and healthy heart tissues, potentially indicating a role in cardiac function or disease pathology. The analysis could, however, not incorporate nanopore long-read data. Hence, the present data underscores the identification of hitherto unrecognized tropomyosin isoforms. *TPM3* gene products are

among the most highly responsive in heart failure, approaching expression levels comparable to the dominant *TPM1* gene products. "In the present study, in vitro motility assays were conducted to investigate the calcium sensitivity of Tpm3.12 and Tpm1.1 and measure filament speed as readout. The results of these assays reveal that Tpm3.12 exhibits greater calcium sensitivity compared to Tpm1.1. In cardiac muscle, increased calcium sensitivity changes important properties of the cardiac contraction cycle, often observed in cardiomyopathies (PMID: 23022395, Asp et al, 2013)." Collectively, our study shows an orchestrated remodeling of the cardiac isoform landscape during heart failure. The changes are highly reproducible and comparable in ischemic and non-ischemic causes. With novel compounds being tested in clinical trials, it becomes more crucial to understand the targeted isoforms

**Table 1. Splicing machinery regulating tropomyosin isoforms in human left ventricle cells.**

| Isoform | Transcript | Primary splicing regulator | Supporting regulators | Splicing complexes and cofactors | Upstream signaling pathways | Disease associations |
|---|---|---|---|---|---|---|
| Tpm1.1 | TPM1-207 | RBM20 | MBNL1, SRSF1, SRSF2, RBFox1 | RBM20-repressed spliceosome, U2 snRNP, PTBP1 | PKC, ERK/MAPK, RBM20-mediated exon repression | Heart failure, hypertrophic cardiomyopathy (HCM), dilated cardiomyopathy (DCM) |
| Tpm3.12 | TPM3-224 | SLM2 | hnRNP A1, SRSF1, CELF1 | hnRNP-enriched splicing complexes, CELF1-dependent spliceosome | SLM2-driven exon inclusion, ERK/MAPK, Calcium signaling | Dilated cardiomyopathy (DCM), cytoskeletal instability |
| Tpm3.13 | TPM3-210 | SLM2 | MBNL1, hnRNP A1 | MBNL1-associated spliceosome, U2AF65 | SLM2-dependent exon switching, TGF-β signaling | Cardiac fibrosis, myofibril disorganization |
| Tpm3.1 | TPM3-212 | RBFox1 | MBNL1, MBNL2 | RBFox1-bound U1 snRNP, UGCAUG-dependent enhancer complex | RhoA/ROCK, Actin cytoskeletal regulation | Vascular remodeling post-MI, altered fibroblast function |
| Tpm3.2 | TPM3-206 | SLM2 | hnRNP A1, SRSF1 | hnRNP A1/U2AF65-mediated splicing machinery | SLM2-mediated exon selection, TGF-β signaling | Fibrotic heart disease, increased ECM deposition |

and their variations under different disease conditions. This might improve efficiency and reduce unwanted side effects. Especially the cardiac sarcomere seems highly flexible regarding its isoform repertoire.

## Limitations

A limitation of this study is the use of bulk RNA sequencing, which captures transcriptomic changes from whole tissue rather than individual cardiomyocytes. While non-myocytes contribute to heart tissue remodeling in heart failure, their influence on transcript abundance is likely limited due to the dominant transcriptional output of cardiomyocytes (Litviňuková et al, 2020). Single-cell and single-nucleus RNA sequencing studies have demonstrated that key sarcomeric transcript changes, including isoform switching in *TPM1* and *TPM3*, are largely preserved across methods, suggesting that the observed shifts primarily reflect cardiomyocyte remodeling (Tabula Sapiens et al, 2022; Koenig et al, 2022). However, bulk sequencing cannot fully resolve cell-type-specific contributions, and future studies using single-cell approaches will be essential to confirm the cardiomyocyte-specific nature of these isoform changes (Cadosch et al, 2024; Koenig et al, 2022; Tabula Sapiens et al, 2022).

## Methods

**Reagents and tools table**

| Reagent/resource | Reference or source | Identifier or catalog number |
|---|---|---|
| **Experimental models** | | |
| Left ventricular heart tissue from DCM and ICM patients | University Hospital Heidelberg | |
| Healthy left ventricles | BioServe/Reprocell | |
| **Chemicals, enzymes and other reagents** | | |
| DNA/RNA/miRNA kit | Qiagen | 80204 |
| Maxima H reverse transcriptase | Thermo Scientific | EP0751 |
| Exonuclease 1 | NEB | M0293 |
| AMPure XP beads | Beckman Coulter | A63882 |
| Ultra II End-prep module | NEB | E7546S |
| **Software** | | |
| DESeq2 | https://bioconductor.org/packages/devel/bioc/html/DESeq2.html | |
| DEXSeq | https://bioconductor.org/packages/devel/bioc/html/DEXSeq.html | |
| DRIMseq | https://bioconductor.org/packages/devel/bioc/html/DRIMSeq.html | |
| SUPPA | https://github.com/comprna/SUPPA | |
| SQANTI | https://github.com/ConesaLab/SQANTI | |

| Reagent/resource | Reference or source | Identifier or catalog number |
|---|---|---|
| **Other** | | |
| GridION flowcell | R9.4.1 | Nanopore |

## Human left ventricular heart tissue

The present study has been approved by the ethics committee of the Medical Faculty of Heidelberg University (appl. no. S-390/2011), and all participants have given written informed consent, and all experiments have been conducted conforming to the principles set out in the WMA Declaration of Helsinki and the Department of Health and Human Services Belmont Report. Control samples were used according to the protected health information (45 C.F.R. 164.514 e2) (bioserve) and the BCI informed consent F-641-5 (biochain). Basic proband phenotypes are listed in Appendix Table 1. The cohort size of included probands was limited by the availability of explanted hearts.

## RNA isolation

Heart pieces were stored at −80 °C, and RNA was extracted from 200 ng tissue using the AllPrep DNA/RNA/miRNA Kit (Qiagen, #80204). New sample IDs, non-indicative of the study group, were introduced to support a blinded analysis. However, we did not explicitly randomize the samples. RNA concentrations were determined on Agilent Technology's Fragment Analyzer with either the standard sensitivity (range 5–500 ng/μL) or the high sensitivity kit (range 50–5000 pg/μL) according to the manufacturer's protocol. RNA quality number (RQN) was measured to assess the quality of the RNA.

## Generation of nanopore cDNA libraries and sequencing

For nanopore sequencing, 30 ng of mRNA was mixed with 0.3 ng spike-in control Lexogen SIRV E0 mix (#050.01). The libraries were prepared according to the manufacturer's protocol for the DCS-109 kit; however, small changes were introduced. In brief, the first step of the library preparation involves cDNA synthesis with a polydT primer (VNP offered by Nanopore) and the Maxima H reverse transcriptase (Thermo Scientific™, #EP0751). Due to maxima Hs terminal transferase activity, it is possible to use the additional bases as an anchoring site for a strand switching primer (SSP, provided by Nanopore) to synthesize the opposite strand as well. Primers complementary to the polydT primer and SSP primer overhangs (provided by Nanopore) were used for the LongAmp® Taq Polymerase PCR. Followed by an exonuclease I (20 units, NEB, #M0293) digestion, PCR products were purified with AMPure XP beads (Beckman Coulter, #A63882). DNA repair to obtain 5'phosphorylated and 3'dA-tailed sites was performed with Ultra II End-prep module from NEB (#E7546S), and DNA was purified with 60 μL AMPure XP beads. The provided nanopore ONT adapters were ligated, and a 60 ng final library was prepared according to the manufacturer's instructions and loaded on a primed GridION flowcell (R9.4.1). Sequencing was performed with one sample per flowcell for 48 h on the GridION, which basecalls directly with Guppy.

## Protein expression and purifications

Tropomyosins were overexpressed by transforming *E. coli* BL21(DE)-pNatB with tropomyosin constructs cloned in either the pET-23a (+) vectors or pETpRha vectors. The pNatB plasmid encodes the regulatory subunit naa25˙ and the catalytic subunit naa20˙ of the NatB complex from fission yeast, both under the control of a T7 promoter (Johnson et al, 2010). Human troponin subunits (cTnC, cTnI, cTnT) were cloned into pET11c vectors containing ampicillin and chloramphenicol resistance and over-expressed in *E.coli* BL21 (RosettaDE3) pLysS cells. Purification of cTnI and cTnC was performed according to (al-Hillawi et al, 1994), while cTnT purification followed the protocol described by (Krüger et al, 2003).

## Quantitative proteomics meta-analysis

We performed a meta-analysis of quantitative proteomics datasets to assess differential expression of tropomyosin isoforms. The analysis included three publicly available studies on human myocardial tissue from heart failure patients and nonfailing controls (Chen et al, 2018; Hunter et al, 2024; Jani et al, 2025). Raw or processed protein quantification data were obtained from the open source repositories. For datasets in which multiple protein groups mapped to the same gene symbol, intensities were aggregated to the gene level by summing the values across all corresponding protein groups on the linear (non-log-transformed) scale. Log2 transformation was applied to intensity values prior to analysis, except for datasets that already provided Log2-transformed values. Differential expression was assessed using linear models to compare heart failure patients versus controls (limma package, v3.58.1), adjusting for age and sex. For meta-analysis, effect sizes (log2 fold changes) and standard errors for each gene were extracted from each study. Random-effects meta-analysis was performed using the metafor package (v4.4-0), with inverse-variance weighting. Model assumptions (normality and homoscedasticity) were checked via residual diagnostics. Genes with data from at least two studies were included in the meta-analysis. Forest plots were generated for key genes.

## In vitro motility assay

Fluorescently labeled actin gliding assays on surface-bound $\beta$-cardiac myosin HMM were conducted as previously described (Reindl et al, 2022) using an Olympus IX70 fluorescent microscope. In brief, 0.2 mg/ml $\beta$-cardiac myosin HMM was diluted in assay buffer (25 mM MOPS, 50 mM KCl, 5 mM MgCl$_2$, 10 mM DTT, pH 7.4) and bound to the cover slip of a flow cell pre-coated using 1% nitrocellulose dissolved in pentyl-acetate. To minimize nonspecific actin binding, the surface was blocked with 0.5 mg/ml bovine serum albumin. F-actin labeled with 20 nM Atto550-Phalloidin (Merck) was incubated on the surface for 3 min under rigor conditions. Motility was initiated by adding 2 mM ATP in an assay buffer supplemented with 0.5% methylcellulose and an anti-photobleaching mix (5 mg/ml glucose, 100 μg/ml glucose-oxidase, 100 μg/ml catalase). At least three video sequences were captured using Olympus xcellence RT software at either one or two frames per second. The velocity and filament length distribution of at least 300 actin filaments were analyzed using FiJi software with a customized version of the wrMTrck plugin (Nussbaum-Krammer et al,

## The paper explained

### Problem

Cardiomyopathy refers to a disease of the heart muscle that involves structural and/or electrical abnormalities. It includes four major types, with dilated cardiomyopathy (DCM) being the most common. Disruptions in alternative RNA splicing—regulated by proteins like RBM20 or SLM2—can interfere with normal gene isoform expression, leading to severe forms of cardiomyopathy in both human patients and animal models. Although new treatments are targeting parts of the heart's sarcomere, the role of isoform switching in heart disease is still not well understood.

### Results

To address this, we used nanopore long-read sequencing to analyze the complete set of RNA transcripts in the left ventricles of 13 healthy individuals, ten patients with DCM, and ten with ischemic cardiomyopathy (ICM). We identified a total of 78,520 transcripts, with 31% representing previously unrecognized isoforms of existing genes. Strikingly, we find for 10/11 sarcomeric genes a significantly dysregulated isoform. As one example, we studied tropomyosin genes and found that TPM1, the main cardiac gene, showed moderate increases in its transcript levels. In contrast, TPM3, which is usually expressed at low levels in a healthy heart, showed a significant increase in failing hearts.

### Impact

Our research highlights a coordinated shift in the expression of heart-specific gene isoforms during heart failure. These changes were consistent across both ischemic and non-ischemic forms of the disease. As new therapies are being developed and tested, gaining a clearer understanding of which isoforms are involved—and how their expression changes under disease conditions—will be increasingly important.

2015; Schindelin et al, 2012). For the measurements involving $Ca^{2+}$ ions, free $Ca^{2+}$ ion concentrations were calculated using MAXCHELATOR with the following parameters: 37 °C, 0.055 N ion contribution, pH = 7.4, ATP = 0.002 M, EGTA = 0.001 M and 0.004 M $Mg^{2+}$

## Data availability

Aggregated data can be accessed from https://zenodo.org/records/17853862 or https://ccb-web.cs.uni-saarland.de/cms .

The source data of this paper are collected in the following database record: biostudies:S-SCDT-10_1038-S44321-025-00370-9.

## Peer review information

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

## Acknowledgements

This study was supported by CRC 1550 (Molecular Circuits of Heart Disease) and the Leducq Foundation (CASTT grant). DM was supported by grants from Deutsche Forschungsgemeinschaft (Project number 462266917) and the German Federal Ministry of Education and Research under Grant Agreement 01GM1922B.

## Author contributions

**Jan Haas**: Formal analysis; Writing—original draft; Writing—review and editing. **Sarah Schudy**: Formal analysis; Investigation; Writing—original draft; Writing—review and editing. **Benedikt Rauscher**: Software; Formal analysis; Visualization. **Ana Muñoz Verdú**: Resources; Software; Writing—review and editing. **Steffen Roßkopf**: Software; Methodology. **Christoph Reich**: Formal analysis; Writing—review and editing. **Gizem Donmez Yalcin**: Writing—original draft; Writing—review and editing. **Abdullah Yalcin**: Investigation; Writing—review and editing. **Timon Seeger**: Resources. **Christoph Dieterich**: Resources. **Manuel H Taft**: Resources; Validation. **Marc Freichel**: Resources. **Dirk Grimm**: Resources; Methodology. **Dietmar Manstein**: Resources; Supervision; Investigation; Writing—original draft; Writing—review and editing. **Johannes Backs**: Resources; Funding acquisition. **Norbert Frey**: Resources; Funding acquisition. **Lars Steinmetz**: Conceptualization; Resources; Supervision; Methodology. **Benjamin Meder**: Conceptualization; Resources; Supervision; Funding acquisition; Writing—original draft; Project administration; Writing—review and editing.

Source data underlying figure panels in this paper may have individual authorship assigned. Where available, figure panel/source data authorship is listed in the following database record: biostudies:S-SCDT-10_1038-S44321-025-00370-9.

## Funding

## Disclosure and competing interests statement

The authors declare no competing interests.

# Expanded View Figures

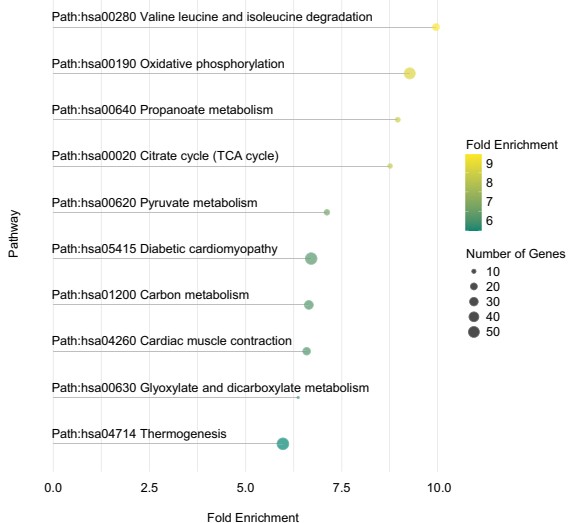

**ShinyGO - Pathway Enrichment Of Upregulated Isoforms (Heart Failure vs.Control)**

**Figure EV1.  Pathway enrichment of upregulated isoforms (heart failure vs. control).**

We performed a pathway analysis (ShinyGO) on upregulated isoforms in heart failure (DCM + ICM) vs. controls. Color scheme visualizes the fold enrichment with darker color being stronger enriched, while the size of the bullet represents the number of shared genes in the respective pathway. Data for running the analysis is provided in Dataset EV1.

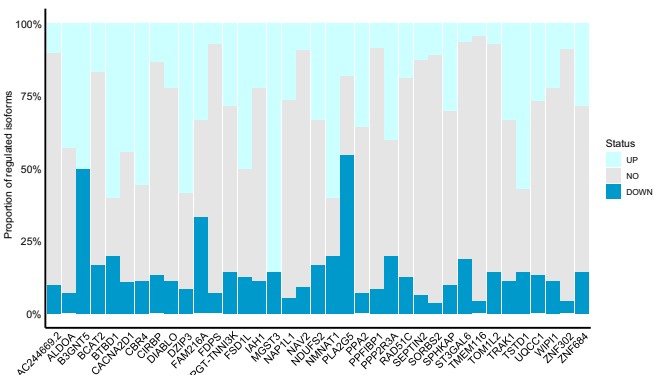

**Figure EV2.  Genes with up- and down-regulated Isoforms (heart failure vs. control).**

Shown are the number of upregulated ("UP"), downregulated ("DOWN"), and not regulated ("NO") per gene for coding genes which show regulated transcript isoforms in opposing directions.

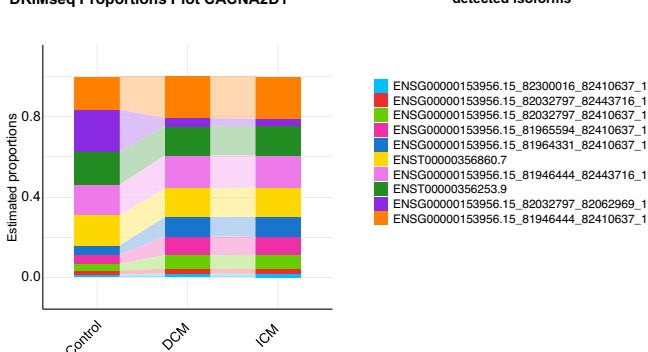

**Figure EV3. DRIMseq proportions plot.**

Overview of detected isoforms for *CACNA2D1* and their proportional contribution to all detected isoforms of the gene.

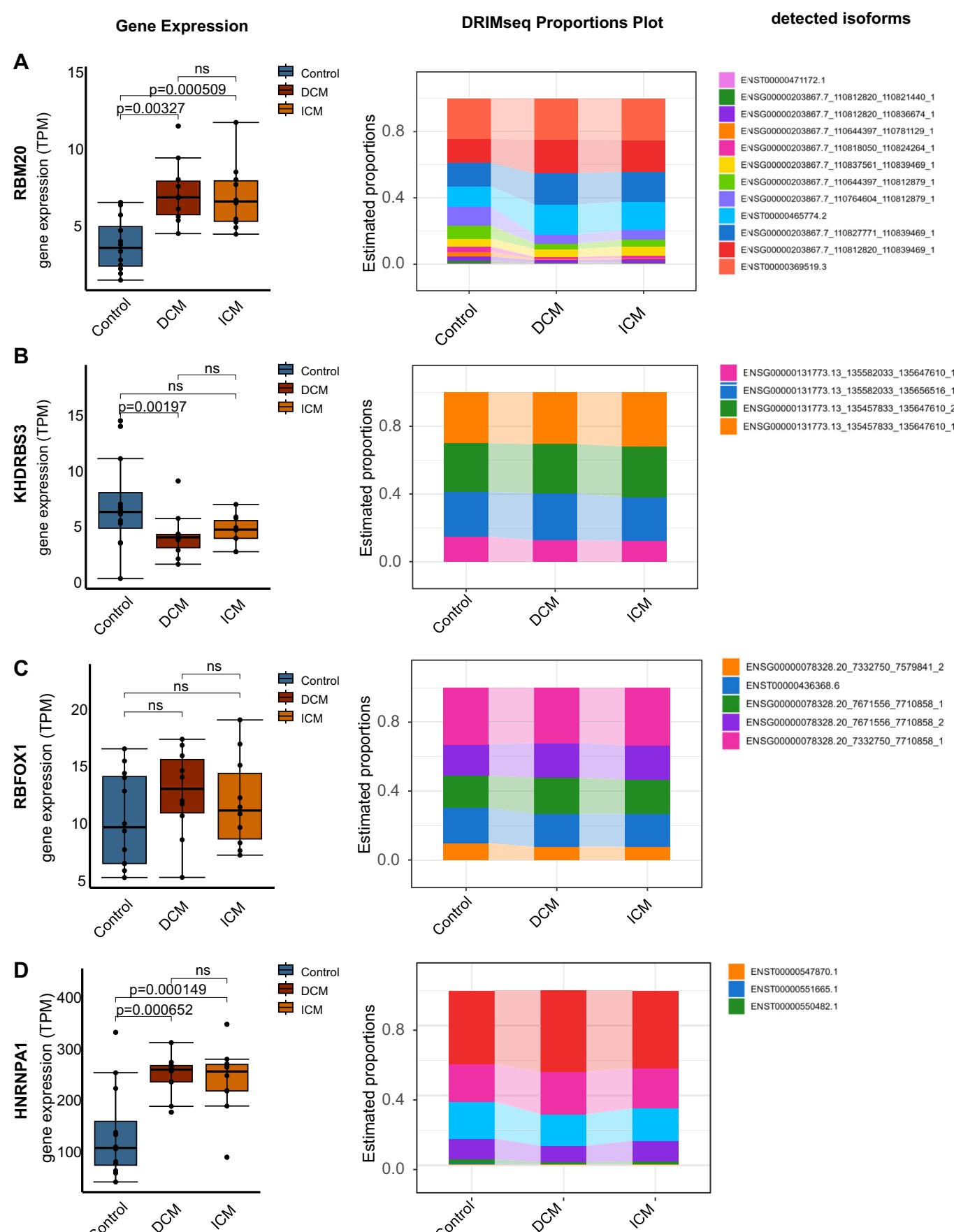

◄ **Figure EV4. Gene expression and DRIMseq proportions plot for important splice factor genes.**

Gene expression in transcript per million (TPM) for CTRL ($n = 13$), DCM ($n = 10$), and ICM ($n = 10$) probes. Box plots in the left panel show gene expression of the splice factors RBM20 (**A**), KHDRBS3 (**B**), RBFOX1 (**C**), and HNRNP A1 (**D**). The right panel displays an overview of the proportion of the detected transcript isoforms of the respective genes, as determined by DRIMseq. Values ranged as following for *RBM20*-CTRL (min = 0.355; max = 6.36; median = 3.19; q1 = 2.05; q3 = 4.55; p95 = 6.32); *RBM20*-DCM (min = 4.34; max = 11.3; median = 6.69; q1 = 5.57; q3 = 7.74; p95 = 11.0); *RBM20*-ICM (min = 4.30; max = 11.6; median = 6.42; q1 = 5.13; q3 = 7.76; p95 = 11.0); *KHDRBS3*-CTRL (min = 0.112; max = 14.3; median = 6.28; q1 = 5.06; q3 = 10.9; p95 = 14.2); *KHDRBS3*-DCM (min = 1.39; max = 8.88; median = 4.07; q1 = 2.88; q3 = 4.07; p95 = 8.27); *KHDRBS3*-ICM (min = 2.52; max = 6.77; median = 4.50; q1 = 3.72; q3 = 5.32; p95 = 6.57); *RBFOX1*-CTRL (min = 5.05; max = 16.3; median = 9.78; q1 = 6.28; q3 = 13.8; p95 = 16.1); *RBFOX1*-DCM (min = 5.07; max = 17.2; median = 12.8; q1 = 10.7; q3 = 15.4; p95 = 17.1); *RBFOX1*-ICM (min = 7.01; max = 18.9; median = 10.9; q1 = 8.43; q3 = 14.2; p95 = 18.5); *HNRNPA1*-CTRL (min = 36.6; max = 329.0; median = 105.0; q1 = 72.6; q3 = 218.0; p95 = 329.0); *HNRNPA1*-DCM (min = 172.0; max = 307.0; median = 255.0; q1 = 231.0; q3 = 263.0; p95 = 301.0); *HNRNPA1*-ICM (min = 84.6; max = 343.0; median = 251.0; q1 = 214.0; q3 = 265.0; p95 = 331.0).

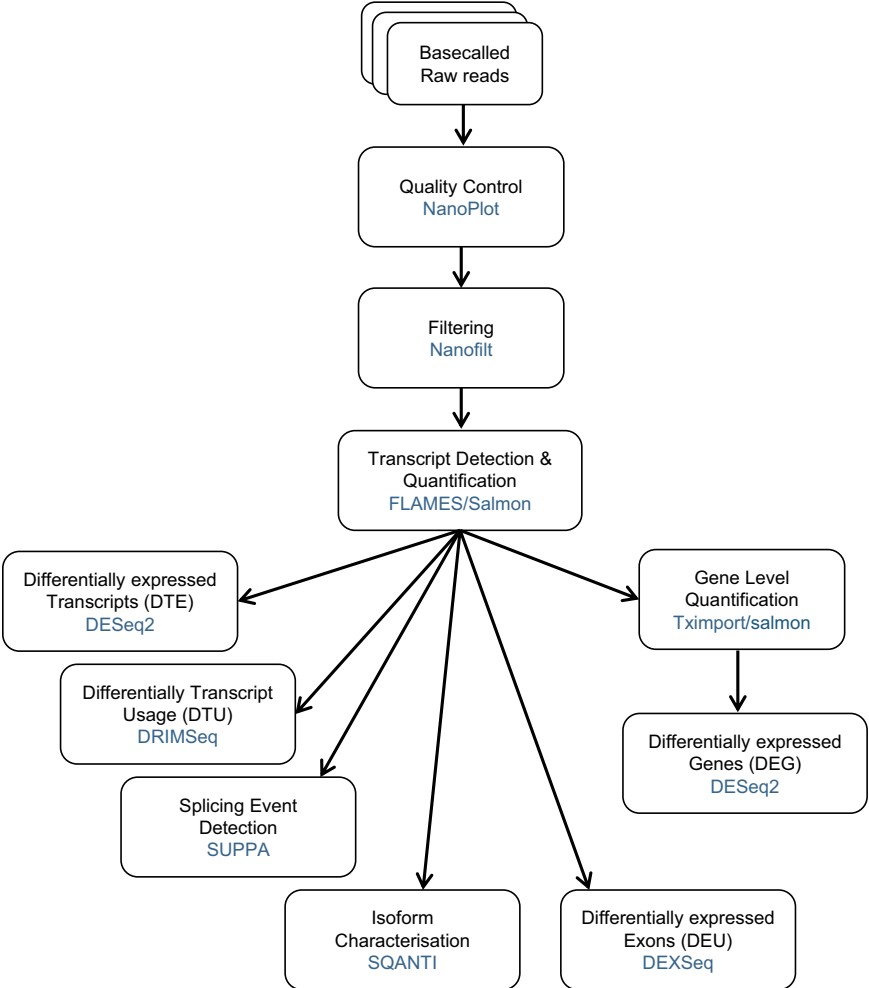

**Figure EV5.   Analysis pipeline.**

Scheme is showing the workflow of the data analysis and list used tools per analysis.

