## [Peer Review File · EMBO Molecular Medicine]

Sarcomeric Remodelling in Human Heart Failure unraveled by single molecule long read sequencing

Jan Haas, Sarah Schudy, Benedikt Rauscher, Ana Munoz, Steffen Roßkopf, Christoph Reich, Gizem Doenmez Yalcin, Abdullah Yalcin, Timon Seeger, Christoph Dieterich, Manuel Taft, Marc Freichel, Dirk Grimm, Dietmar Manstein, Johannes Backs, Norbert Frey, Lars Steinmetz, and Benjamin Meder

Corresponding author: Benjamin Meder (benjamin.meder@med.uni-heidelberg.de)

Review Timeline:

Submission Date:	28th Feb 25
Editorial Decision:	1st Apr 25
Revision Received:	16th Sep 25
Editorial Decision:	9th Oct 25
Revision Received:	6th Nov 25
Editorial Decision:	13th Nov 25
Revision Received:	11th Dec 25
Accepted:	12th Dec 25

Editor: Lise Roth

Transaction Report:

1st Apr 2025

Dear Prof. Meder,

Thank you for the submission of your manuscript to EMBO Molecular Medicine. We have now received feedback from the three reviewers who agreed to evaluate your manuscript. As you will see from the reports below, the referees acknowledge the interest of the study and are overall supporting publication of your work pending appropriate revisions.

Addressing the reviewers' concerns in full will be necessary for further considering the manuscript in our journal, and acceptance of the manuscript will entail a second round of review. However, please note that we do NOT ask for validation of the data in an animal in vivo model (point raised by ref #3), but rather for in vitro investigation to strengthen the mechanistic understanding.

EMBO Molecular Medicine encourages a single round of revision only and therefore, acceptance or rejection of the manuscript will depend on the completeness of your responses included in the next, final version of the manuscript. For this reason, and to save you frustration at the end, I would strongly discourage you from returning an incomplete revision.

We are expecting your revised manuscript within three to four months, if you anticipate any delay, please contact us.

We require:

4) A .docx formatted letter INCLUDING the reviewers' reports and your detailed point-by-point responses to their comments. As part of the EMBO Press transparent editorial process, the point-by-point response is part of the Review Process File (RPF), which will be published alongside your paper.

5) A complete author checklist, which you can download from our author guidelines (<https://www.embopress.org/page/journal/17574684/authorguide#submissionofrevisions>). Please insert information in the checklist that is also reflected in the manuscript. The completed author checklist will also be part of the RPF.

6) All Materials and Methods need to be described in the main text using our 'Structured Methods' format. According to this format, the Methods section includes a Reagents and Tools Table (listing key reagents, experimental models, software and relevant equipment and including their sources and relevant identifiers) followed by a Methods and Protocols section describing the methods, ideally using a step-by-step protocol format. The aim is to facilitate adoption of the methodologies across labs. Please download and fill our Reagents and Tools Table template (.docx), which you can find in our author guidelines: <https://www.embopress.org/page/journal/14693178/authorguide#structuredmethods>.

<https://www.embopress.org/doi/10.15252/msb.20178071>

7) Please note that all corresponding authors are required to supply an ORCID ID for their name upon submission of a revised manuscript.

8) It is mandatory to include a 'Data Availability' section after the Materials and Methods. Before submitting your revision, primary datasets produced in this study need to be deposited in an appropriate public database, and the accession numbers and database listed under 'Data Availability'. Please remember to provide a reviewer password if the datasets are not yet public (see

<https://www.embopress.org/page/journal/17574684/authorguide#dataavailability>).

9) For data quantification: please specify the name of the statistical test used to generate error bars and P values, the number (n) of independent experiments (specify technical or biological replicates) underlying each data point and the test used to calculate p-values in each figure legend. The figure legends should contain a basic description of n, P and the test applied. Graphs must include a description of the bars and the error bars (s.d., s.e.m.). Please provide exact p values.

10) Our journal encourages inclusion of *data citations in the reference list* to directly cite datasets that were re-used and obtained from public databases. Data citations in the article text are distinct from normal bibliographical citations and should directly link to the database records from which the data can be accessed. In the main text, data citations are formatted as follows: "Data ref: Smith et al, 2001" or "Data ref: NCBI Sequence Read Archive PRJNA342805, 2017". In the Reference list, data citations must be labeled with "[DATASET]". A data reference must provide the database name, accession number/identifiers and a resolvable link to the landing page from which the data can be accessed at the end of the reference. Further instructions are available at .

11) We replaced Supplementary Information with Expanded View (EV) Figures and Tables that are collapsible/expandable online. EV Figures should be cited as 'Figure EV1, Figure EV2' etc... in the text and their respective legends should be included in the main text after the legends of regular figures.

12) The paper explained: EMBO Molecular Medicine articles are accompanied by a summary of the articles to emphasize the major findings in the paper and their medical implications for the non-specialist reader. Please provide a draft summary of your article highlighting

13) Author contributions: CRediT has replaced the traditional author contributions section because it offers a systematic machine readable author contributions format that allows for more effective research assessment. Please remove the Authors Contributions from the manuscript and use the free text boxes beneath each contributing author's name in our system to add specific details on the author's contribution. More information is available in our guide to authors.

Please also suggest a visual abstract to illustrate your article as a PNG file 550 px wide x 300-600 px high. A cropped portion of this image will serve as thumbnail for the table of content on our webpage.

16) As part of the EMBO Publications transparent editorial process initiative (see our Editorial at <http://embomolmed.embopress.org/content/2/9/329>), EMBO Molecular Medicine will publish online a Review Process File (RPF) to accompany accepted manuscripts.

In the event of acceptance, this file will be published in conjunction with your paper and will include the anonymous referee reports, your point-by-point response and all pertinent correspondence relating to the manuscript. Let us know whether you agree with the publication of the RPF and as here, if you want to remove or not any figures from it prior to publication.

I look forward to receiving your revised manuscript.

Yours sincerely,

Lise Roth

**** Reviewer's comments ****

Referee #1 (Remarks for Author):

In this study, Haas et al. studied sarcomeric remodeling during heart failure using long-read sequencing of left ventricular tissue from DCM and ICM patients. Results revealed a significant shift in transcript isoforms, with 31% of the detected 78,520 transcripts being novel. The authors claim that despite differing etiologies, dilated and ischemic cardiomyopathy exhibited a largely convergent isoform landscape. Focusing on individual genes, the authors show increased expression levels of TPM1-207 and TPM3-224 in both disease groups. Because TPM3-224 exhibits increased calcium sensitivity compared to TPM1-207, they suggest it presents a potential adaptation to contractile demands in heart failure and emphasize the importance of understanding isoform-specific alterations for future therapeutic strategies. Overall, it is an interesting study which would benefit from more detailed analysis of isoform differences.

Major comments:

- There is no comprehensive overview of up- and downregulated isoforms, just a volcano plot with a few annotated individual genes. The study would benefit from categorizing transcripts and some sort of GOterm/KEGG analysis: Is there up- and downregulation of myofilament proteins, splice factors etc? A few individual transcripts are discussed, but overall trends would be interesting. Changes in major sarcomeric proteins are shown (Figure 4e), but detailed isoform changes are not shown or discussed. There is also no overall data file showing individual transcript changes.
- Considering that long-read sequencing is one of the novel aspects of this study: Are there other examples where specific isoforms of a gene are upregulated / downregulated but others are not or go in the opposite direction? This would give us more insight into how splicing helps to adapt to contractile demands during heart failure. Existing examples of transcriptomic changes in the discussion might be interesting, but are not a unique aspect of this study and have been found in short-read data.
- The authors mention detecting 31% new isoforms among their detected transcripts. The manuscript is lacking detailed information about bioinformatic methods (how is transcript characterized as "novel"?). Gencode has different annotated versions (comprehensive or basic) that can be used for analysis. Which one was used for this study?
- Tpm3.12 is shown to exhibit greater calcium sensitivity than Tpm1.1. However, several studies have demonstrated that overexpression of TPM3 in mouse hearts leads to decreased myofilament calcium sensitivity. How could this discrepancy be explained? Is it specifically related to the exons included in TPM3-224?
- What exactly is encoded by exons that are differentially included between the four shown TPM isoforms? TPM3-224 is chosen for additional analysis without further explanation. It should be discussed how protein domains included in different isoforms might impact function, especially considering discrepancies to published TPM3 data.

Minor comments:

- I understand the reasoning for different nomenclature for the studied isoforms, but tropomyosin names should be unified or mentioned both where necessary in the text and figures: Tpm1.1 (TPM1-207, Tpm3.12 (TPM3-224). This would make it easier to compare figures 6 and 7 and portions of the text.
- As mentioned above, some figures/legends would benefit from more detailed information (for example GENCODE: which version in figure 2B)

Referee #2 (Remarks for Author):

This initial manuscript submission by Haas and colleagues details the interesting investigation of differential long read RNA sequencing in normal and cardiomyopathic human hearts. The authors utilize nanopore sequencing technology to obtain long sequencing reads that allowed for the detection of splicing differences in key cardiomyocyte genes. The authors detail specific splice variants/isoforms of the tropomyosin TPM3, which is mechanistically followed up by protein expression and motility experiments. Overall, the study is very interesting, well conducted, and concisely presented. A couple of concerns exist, however:

- 1) Throughout the manuscript, the authors confuse the alpha-MHC (MYH6) as being the predominant myosin isoform in adult human healthy ventricles. While that is correct for rodents, the beta-MHC isoform (MYH7) is predominantly expressed in human hearts. All instances of this mistake should be corrected. Maybe providing the gene symbol with the discussion would help.
- 2) The sliding filament experiment is very nice addition to this manuscript, as it provides mechanistic insight for this novel tropomyosin isoform. The results section for this experiment, however, comes to an abrupt end that provides no interpretation of the data. The discussion also does not go much into the physiological importance of these findings. It is suggested that these areas are expanded on to highlight the potential significance of this novel isoform. Also, representative images and/or videos of the experiments would be helpful for reader interpretation.

Referee #3 (Comments on Novelty/Model System for Author):

This study provides important information related to the switch in transcript isoforms, specifically related to sarcomere structure in heart failure patients. However, the majority of the data is computational analysis of the RNA samples and requires further support in terms of RNA/protein expression data. The story would be strengthened if the data were verified in an animal in vivo model. There are some other design and technical that should be considered which I detailed below. Overall this is very interesting study that would be of interest to the journal readership.

Referee #3 (Remarks for Author):

Summary

In this study, the authors performed nanopore long-read sequencing to map the full-length transcriptome of left ventricular tissue from healthy controls and patients with dilated cardiomyopathy (DCM) and ischemic cardiomyopathy (ICM). They identified several previously known and novel transcript isoforms of genes. Interestingly, similar trends in transcriptomes of DCM and ICM were observed. Both heart failure cases (DCM and ICM) displayed isoform shift in sarcomere genes, such as upregulation of transcript isoforms of tropomyosin genes TPM1 and TPM3.

Comments

This study provides important information related to the switch in transcript isoforms, specifically related to sarcomere structure in heart failure patients. However, the majority of the data is computational analysis of the RNA samples and requires further support in terms of RNA/protein expression data.

Specific points

1. Based on the data provided here, the transcriptomes of DCM and ICM are quite different from the non-failing heart controls. The transcriptomes are regulated by splicing factors, cofactors, and upstream signaling pathways (data provided in Table 1). Does the change in transcript isoforms of Sarcomeric genes (including Tropomyosin isoforms) relate to alterations in these factors (RBM20, SLM2, hnRNP A1, and RBFOX1)? The outcome of this study will identify the targets to prevent the sarcomeric pathological remodeling and structural defects in heart failure patients.
2. The authors should provide the gene and protein expression of tropomyosin isoforms in the healthy and heart failure patients tested in this study.
3. The authors should test the tropomyosin isoforms on contractility and function in the cardiac myocytes under stress and physiological conditions.
4. The authors should also test the other top upregulated and downregulated genes in these patients.

Point-by-point response

Referee #1:

In this study, Haas et al. studied sarcomeric remodeling during heart failure using long-read sequencing of left ventricular tissue from DCM and ICM patients. Results revealed a significant shift in transcript isoforms, with 31% of the detected 78,520 transcripts being novel. The authors claim that despite differing etiologies, dilated and ischemic cardiomyopathy exhibited a largely convergent isoform landscape. Focusing on individual genes, the authors show increased expression levels of TPM1-207 and TPM3-224 in both disease groups. Because TPM3-224 exhibits increased calcium sensitivity compared to TPM1-207, they suggest it presents a potential adaptation to contractile demands in heart failure and emphasize the importance of understanding isoform-specific alterations for future therapeutic strategies. Overall, it is an interesting study which would benefit from more detailed analysis of isoform differences.

Major comments:

- There is no comprehensive overview of up- and downregulated isoforms, just a volcano plot with a few annotated individual genes.

As pointed out by the reviewer, the volcano-plot is more tailored for a general overview on the data. Based on the reviewer's suggestion and to improve the value of the study for the reader, we have now added supplemental data tables with all significantly up or downregulated transcript-isoforms in heart failure vs. controls. In response to other points, we now also provide new figures providing insight into other genes. (Tables EV2, EV3, Figure EV2-EV4).

- The study would benefit from categorizing transcripts and some sort of GOterm/KEGG analysis: Is there up- and downregulation of myofilament proteins, splice factors etc?

We agree with the reviewer that functional categorization of the altered isoforms can further contribute to a better understanding on the functional changes undergoing in heart failure. From the supplemental data table 3, we now have subjected every gene with at least one significantly different isoform to a pathway analysis using shinyGO (v.0.82). We have included results of this analysis in Figure EV1. The main text describing those results now reads:

"Next, we performed a pathway analysis on upregulated isoforms in heart failure vs. controls. As expected, cardiomyopathy pathways are significantly being enriched, e.g. "Dilated cardiomyopathy" (Enrichment FDR = 6.08e-05; Fold Enrichment = 3.98) or "Hypertrophic cardiomyopathy" (Enrichment FDR = 0.006; Fold Enrichment = 2.99), this is somehow expected and not surprising and yields little novel insight into pathophysiology of heart failure. However, there are several pathways that are highly enriched and carry many genes in the pathways related to energy production metabolism. This finding is in line with our previous data on the dysregulation of several metabolites involved in glycolysis and citric acid cycle (Haas et al, 2021). The importance of such pathways has also recently been reviewed e.g. by Lopaschuk et al. (Lopaschuk et al, 2021)."

We also followed the reviewer's advice and provide now individual gene and isoform expression plots for the discussed splice factors (Figure EV4). As shown, *RBM20* and *hnRNPA1* display changes in expression of isoforms.

- A few individual transcripts are discussed, but overall trends would be interesting. Changes in major sarcomeric proteins are shown (Figure 4e), but detailed isoform changes are not shown or discussed. There is also no overall data file showing individual transcript changes.

As mentioned above, we now provide detailed supplemental data for heart failure vs. Controls (Tables EV2 and EV3) increasing the transparency.

- Considering that long-read sequencing is one of the novel aspects of this study: Are there other examples where specific isoforms of a gene are upregulated / downregulated but others are not or go in the opposite direction? This would give us more insight into how splicing helps to adapt to

contractile demands during heart failure. Existing examples of transcriptomic changes in the discussion might be interesting, but are not a unique aspect of this study and have been found in short-read data.

We thank the reviewer for this valuable suggestion. We have investigated this question and after filtering for genes showing at least one coding transcript being upregulated and another downregulated, we find 37 genes with such changes. We have included it in Figure EV2 and Table EV4.

- The authors mention detecting 31% new isoforms among their detected transcripts. The manuscript is lacking detailed information about bioinformatic methods (how is transcript characterized as "novel"?). Gencode has different annotated versions (comprehensive or basic) that can be used for analysis. Which one was used for this study?

We have used the comprehensive GENCODE gene annotation (.v33). We have added this information to the Figure Legend 2B. Further, an overview on the bioinformatics analysis pipeline has been added as Figure EV1-4. In short, we describe the use of the established FLAMES-pipeline for transcript detection and quantification. For differentially expressed genes, transcripts and exons, we used DESeq2 and DEXSeq. For differential transcript usage we relied on DRIMSeq, SUPPA and SQANTI for splicing detection and isoform characterization.

- Tpm3.12 is shown to exhibit greater calcium sensitivity than Tpm1.1. However, several studies have demonstrated that overexpression of TPM3 in mouse hearts leads to decreased myofilament calcium sensitivity. How could this discrepancy be explained? Is it specifically related to the exons included in TPM3-224?

The study the reviewer is likely referring to is Pieples et al., 2002 (PMID: 12234784), "*Tropomyosin 3 expression leads to hypercontractility and attenuates myofilament length-dependent Ca²⁺ activation.*" In that work, overexpression of TPM3 in transgenic mouse hearts resulted in a reduced Ca²⁺ sensitivity of force generation in detergent-extracted fiber bundles. The authors attributed this to a **gene-level isoform** shift from TPM1 (α -TPM) to TPM3. Importantly, the study did not address **splicing isoform-specific** functional differences. The TPM3 cDNA used was full-length, but the exact splicing isoform cannot be determined from their methods, as their amplification primers were not splice-isoform-specific.

In contrast, our study directly compares **defined splicing isoforms**, Tpm3.12 (TPM3-224) and Tpm1.1 (TPM1-207), and measures Ca²⁺ sensitivity using an in vitro motility assay. The apparent difference in direction of effect between the two studies is also explained by the different functional readouts: Pieples et al. measured the Ca²⁺–force relationship and maximal tension (pCa 4.5), while we assessed unloaded sliding velocity. Because force and shortening velocity are inversely related in muscle fibers (Lieber 2002; Enoka 2008), differences in the measured parameter can appear as opposite trends in Ca²⁺ sensitivity.

Regarding the reviewer's question on exon composition: TPM3-224 and TPM1-207 differ in several non-overlapping potential post-translational modification (PTM) sites (Appendix Figures S1 and S2). Specifically, Tpm3.12 lacks seven putative/proven serine/threonine phosphorylation sites present in Tpm1.1, but has three additional serine/threonine residues at other positions. Such sequence and PTM-site differences—arising from the alternative exons included in each isoform—could plausibly underlie the observed differences in Ca²⁺ sensitivity between Tpm3.12 and Tpm1.1.

- What exactly is encoded by exons that are differentially included between the four shown TPM isoforms? TPM3-224 is chosen for additional analysis without further explanation.

As shown in Figure 6B, the isoform shift toward TPM3-224 was among the largest in terms of fold change. This was the main reason we selected TPM3-224 for further functional analysis. We now provide exon alignments in Appendix Figures S1 and S2.

In both DCM and ICM, TPM3-224 and TPM3-210 transcripts increased, whereas TPM3-206 and TPM3-212 showed no significant change compared to controls. TPM3-224 and TPM3-210 include exon 9a and exons 2b + 1a, whereas TPM3-206 and TPM3-212 contain exon 9d instead. The sequences of exon 9a and exon 9d are highly divergent, encoding distinct amino acids that could confer different post-translational modification (PTM) potentials, although no direct functional annotations are available.

Exons 2b and 1a also show notable sequence characteristics, containing a relatively high proportion of both basic and acidic residues, which may influence tropomyosin's interaction with actin or regulatory proteins. Taken together, the exon usage pattern suggests a disease-associated preference for exon 9a and 2b + 1a in TPM3 transcripts, which may underlie functional differences between isoforms.

- It should be discussed how protein domains included in different isoforms might impact function, especially considering discrepancies to published TPM3 data.

We discussed this point in the revised manuscript.

Minor comments:

- I understand the reasoning for different nomenclature for the studied isoforms, but tropomyosin names should be unified or mentioned both where necessary in the text and figures: Tpm1.1 (TPM1-207, Tpm3.12 (TPM3-224). This would make it easier to compare figures 6 and 7 and portions of the text.

We have now mentioned both tropomyosin names where necessary in the text and figures as the reviewer suggested.

- As mentioned above, some figures/legends would benefit from more detailed information (for example GENCODE: which version in figure 2B)

Please see previous comment, we have added this information to the Figure legend 2B.

Referee #2

This initial manuscript submission by Haas and colleagues details the interesting investigation of differential long read RNA sequencing in normal and cardiomyopathic human hearts. The authors utilize nanopore sequencing technology to obtain long sequencing reads that allowed for the detection of splicing differences in key cardiomyocyte genes. The authors detail specific splice variants/isoforms of the tropomyosin TPM3, which is mechanistically followed up by protein expression and motility experiments. Overall, the study is very interesting, well conducted, and concisely presented. A couple of concerns exist, however:

- Throughout the manuscript, the authors confuse the alpha-MHC (MYH6) as being the predominant myosin isoform in adult human healthy ventricles. While that is correct for rodents, the beta-MHC isoform (MYH7) is predominantly expressed in human hearts. All instances of this mistake should be corrected. Maybe providing the gene symbol with the discussion would help.

We thank the reviewer for this comment, and we have now corrected this mistake throughout the manuscript.

- The sliding filament experiment is very nice addition to this manuscript, as it provides mechanistic insight for this novel tropomyosin isoform. The results section for this experiment, however, comes to an abrupt end that provides no interpretation of the data. The discussion also does not go much into the physiological importance of these findings. It is suggested that these areas are expanded on to highlight the potential significance of this novel isoform. Also, representative images and/or videos of the experiments would be helpful for reader interpretation.

We completely agree that additional visualization such as videos will support a better understanding for the reader. We have now included Supplemental videos 1-6, showing the increased velocity for Tpm3.12 (TPM3-224) (Supplemental Videos 1-3) in comparison to Tpm1.1 (TPM1-207) (Supplemental Videos 4-6). In addition to this, we have also updated the discussion. The respective part now reads:

“In the present study, in vitro motility assays were conducted to investigate the calcium sensitivity of Tpm3.12 and Tpm1.1 and measure filament speed as readout. The results of these assays reveal that Tpm3.12 exhibits greater calcium sensitivity compared to Tpm1.1. In cardiac muscle, increased calcium sensitivity changes important properties of the cardiac contraction cycle, often observed in cardiomyopathies (PMID: 23022395, Asp et al. 2014).”

Referee #3

This study provides important information related to the switch in transcript isoforms, specifically related to sarcomere structure in heart failure patients. However, the majority of the data is computational analysis of the RNA samples and requires further support in terms of RNA/protein expression data. The story would be strengthened if the data were verified in an animal *in vivo* model. There are some other design and technical that should be considered which I detailed below. Overall this is very interesting study that would be of interest to the journal readership. ... This study provides important information related to the switch in transcript isoforms, specifically related to sarcomere structure in heart failure patients. However, the majority of the data is computational analysis of the RNA samples and requires further support in terms of RNA/protein expression data.

Specific points :

- Based on the data provided here, the transcriptomes of DCM and ICM are quite different from the non-failing heart controls. The transcriptomes are regulated by splicing factors, cofactors, and upstream signaling pathways (data provided in Table 1). Does the change in transcript isoforms of Sarcomeric genes (including Tropomyosin isoforms) relate to alterations in these factors (RBM20, SLM2, hnRNP A1, and RBFOX1)? The outcome of this study will identify the targets to prevent the sarcomeric pathological remodeling and structural defects in heart failure patients.

The reviewer is raising an important question, which has also been addressed similarly by reviewer 1. Hence, we now provide visualization for gene expression together with an overview on differential transcript usage for such genes (see Expanded View Figure 4). However, drawing direct conclusions on the individual contributions is difficult.

- The authors should provide the gene and protein expression of tropomyosin isoforms in the healthy and heart failure patients tested in this study.

We agree that isoform-specific protein quantification would be valuable. However, TPM3-210 and TPM3-224, the isoforms overexpressed in DCM and ICM, share >90% sequence identity with TPM1.1 (Appendix), making them indistinguishable with currently available antibodies. The only TPM3-specific antibodies (e.g., LC1 and CG3; Schevzov et al., 2011) target exon 1b, which is unique to TPM3-206 and TPM3-212 — isoforms that do not increase in our transcript data.

To explore potential translational consequences, we performed a meta-analysis of three proteomic studies (Chen et al., Hunter et al., Jani et al.) comparing heart failure (HF) and control samples. This analysis (new panels in Figure 6) shows TPM3 protein to be significantly upregulated in heart failure (pooled log₂ fold change = 0.33 (equals 1.3 fold increase), 95% CI: 0.10–0.56; p = 0.004; I² = 0%), in line with our transcript results. TPM1 showed no consistent difference (random-effects estimate = –0.08, 95% CI: –0.54 to 0.38; p = 0.73; I² = 62.3%). Sample sizes (HF+ vs. HF–) were: Chen et al. (23 vs. 11), Hunter et al. (57 vs. 20), Jani et al. (10 vs. 10).

In addition, sequence analysis revealed that Tpm3.12 (TPM3-224) lacks seven putative/proven serine/threonine phosphorylation sites present in Tpm1.1, but contains three alternative sites, which could influence isoform-specific post-translational regulation.

- The authors should test the tropomyosin isoforms on contractility and function in the cardiac myocytes under stress and physiological conditions.

Our model is a pure *in vitro* system where stress cannot be applied, because of missing components e.g. surface receptors (beta receptor) or kinases (e.g. PKA). The physiological conditions are represented with the physiological calcium concentration. It would require generation of transgenic animals or cells to represent a real physiological condition, however; building such transgenic model systems will definitely need a longer period and would go beyond the scope of the current paper as also stated by the editor.

- The authors should also test the other top upregulated and downregulated genes in these patients.

We thank the referee for this idea and did to perform a genome-wide, proteome-wide investigation. For this purpose, we retrieved heart failure versus control mass-spectrometry studies that were very recently published. Next, we performed a 9-sector analysis of changes in RNA expression versus changes in protein expression under the influence of heart failure. As shown, we can find an established distribution of both fold-changes, showing unregulated, co-regulated and anti-regulated expression changes (Figure

6). For instance, Tropomyosin-3, which we also functionally investigated, shows a concordant upregulation on RNA and protein level (Fig. 6D). We also provide a table for all other concordant expression changes (Table EV5) and updated the results text:

“To assess, whether increased TPM3 transcript levels result in elevated protein levels of this gene, we have performed a meta-analysis on recently published studies based on mass-spectrometry protein quantification (Chen et al, 2018; Hunter et al, 2024; Jani et al, 2025). Significantly higher protein levels in heart failure patients were found for TPM3 (Log2-FC = 0.33 [0.10,0.56]), which means a 26% higher TPM3 expression in HF, confirming our transcript-based findings (Fig. 6D). For TPM1, the analyzed studies showed quite heterogeneous protein expression (Log2-FC = -0.08 [-0.54,0.38]) (Fig. 6E).

To explore the transcriptome-wide and proteome-wide correlation, we conducted a nine-sector analysis which reflects well known mechanisms of mRNA-to-protein correlations (Wang et al, 2019). As shown in Fig. 6F, we find non-regulated (sector 5) as well as concordant (3, 7) and discordant (1, 9) candidates. In total, 527 of 4833 (11%) analyzed genes/proteins were concordantly regulated as consequence of heart failure, including TPM3 (sector 3).” (Fig. 6F).

9th Oct 2025

Dear Prof. Meder,

Thank you for submitting your revised study. We have now received the report from referee #1, who also reviewed your responses to referee #3. As you will see below, this referee is satisfied with the revisions, and I will therefore be able to accept your manuscript once the following editorial concerns are addressed:

1/ Manuscript text:

- Please remove the green highlights and only keep in track changes mode any new modification in the text.
- Please provide up to 5 keywords.
- "Materials and Methods" should be renamed "Methods":
 - o Human samples: please include the full statement that informed consent was obtained from all subjects and that the experiments conformed to the principles set out in the WMA Declaration of Helsinki and the Department of Health and Human Services Belmont Report. If collected and within the bounds of privacy constraints report on age, sex and gender or ethnicity for all study participants.
 - o Statistics: Please provide information on sample size, randomization, blinding and inclusion/exclusion criteria.
- Data Availability section: Please remove "All data is available upon request". This section should list the primary datasets produced in this study, deposited in an appropriate public database, and list the accession numbers and database (with URLs). Please fill in the checklist accordingly.
- Acknowledgements: the information provided should match the information provided in the submission system. Please check whether O1GM1922B should be added to the list of funders in the system.
- Please place the References before the Figure Legends.

2/ Figures:

- Figures and EV Figures should be uploaded as individual, high resolution figure files.
- Please rename the corresponding figure "Figure EV1" - EV5 and add the legends to the manuscript text, after Table 1 and under the heading "Expanded View Figure Legends".
- Please make sure all figures and figure panels are referenced in the text. Currently, a callout is missing for Fig. EV4 and for Movie EV1-EV6. Please correct the callouts to "Appendix Figure S1" and "Appendix Figure S2".
- Please upload Datasets EV1-5 as individual files, i.e. one file per dataset. Please add a legend with the title and a short description to each dataset, in a separate tab/worksheet. Please copy the movie legends to a simple text or word file and zip each legend with the corresponding movie file.
- Appendix: Please add page numbers to the table of contents and the figure legends to the appendix file, underneath the corresponding figure.
- Please address the queries from our data editors in the figure legends:
 1. Please note that the legend for figures EV 1-4 is missing in the manuscript. This needs to be rectified.
 2. Please note that the exact p values are not provided in the legends of figures 6B, EV4 A, D.
 3. Please indicate the statistical test used for data analysis in the legends of figures 1A, B; 4A, B; 5B, EV4 A-D.
 4. Please note that the box plots need to be defined in terms of minima, maxima, centre, bounds of box and whiskers, and percentile in the legends of figures 3B, 5A, 6B, EV4 A-D.
 5. Please note that information related to n is missing in the legends of figures 1A, B; 3B, 4A, B; 5A, 6B, 7B; EV4 A-D.
 6. Please note that scale bar and its definition are missing for figure 7A.

3/ Source Data:

Please only provide numerical data, images and blots underlying figures and figure panels. When figures or figures panels are supported by deposited large scale data, there is no need to provide anything else.

4/ Checklist:

- Please fill in the manuscript and author information (top left corner)
- Antibodies: please fill in the left column and provide antibodies information in the reagents table.
- Please check the section "Laboratory protocol" and whether you need to fill it.
- You indicated that information on Experimental study design and Statistics are provided in the Reagents Table, please check.
- Please correct the Data Availability section (left and right column).

5/ Please include the paper explained in the main manuscript text file.

6/ Thank you for providing a nice visual abstract. Please upload it as an individual png/tiff/jpeg file 550 px wide x 300-600 px high, and make sure that the text remains legible. A cropped portion of this image will serve as thumbnail for the table of content on our webpage.

Please also provide a synopsis text, that should include a short stand first (maximum of 300 characters, including space) as well

as 2-5 one-sentences bullet points that summarizes the paper (maximum of 30 words / bullet point).

7/ As part of the EMBO Publications transparent editorial process initiative (see our Editorial at <http://embomolmed.embopress.org/content/2/9/329>), EMBO Molecular Medicine will publish online a Review Process File (RPF) to accompany accepted manuscripts.

This file will be published in conjunction with your paper and will include the anonymous referee reports, your point-by-point response and all pertinent correspondence relating to the manuscript. Let us know whether you agree with the publication of the RPF.

I look forward to receiving your revised manuscript.

Yours sincerely,

Lise Roth

***** Reviewer's comments *****

Referee #1 (Remarks for Author):

Thank you for the detailed response and the revised manuscript. Major comments regarding the transparency of results (additional information about up- and downregulated isoforms and functional categorization) have been adequately addressed with additional tables and figures. New analysis of isoform changes in opposite directions (Figure EV2) increases the value for the reader. More methodical detailed about the bioinformatic pipeline were added and questions about tropomyosin isoforms were sufficiently discussed. Overall, the revised manuscript meets the criteria for publication in technical quality, clarity, and significance for the field.

The authors addressed the editorial issues.

13th Nov 2025

Dear Prof. Meder,

Thank you for submitting your revised files. I have checked everything, and will be able to accept your manuscript once these remaining minor concerns will be addressed:

1/ Data Availability:

It is currently unclear whether the primary datasets have been deposited, or had been previously deposited in the repository accessible with the provided link. Please clarify if this is a public access-controlled repository that follows ethical obligations to the patients and if it is curated.

2/ Please correct the callouts to movies EV1-6 in the manuscript text.

3/ Source Data:

As mentioned previously, only numerical data, images and blots should be uploaded there. Please remove the other files, and annotate the image provided for Fig. 7.

4/ Checklist:

- Human research participants: you indicated that the information is provided in the Data Availability Section, please check and correct if needed (Appendix Table S1?).
- I could not find the DOI or citation details for external step-by-step protocols, please check and clarify (section "Laboratory Protocol").
- Experimental study design and statistics: please check that the information provided matches the manuscript text (for instance, I could not find information on blinding in the text).

5/ We tried to resize your visual abstract to 550 px wide x 300-600 px high, however the text was not legible anymore. Could you please provide an alternative image at the right dimensions and format (tiff/jpeg/png)? A cropped portion of this image will serve as thumbnail for the table of content on our webpage.

6/ I introduced minor edits in your synopsis, please let me know if you agree or amend as you see fit:

"Cardiomyopathy is a disease of the heart muscle involving structural and/or electrical abnormalities. There are four main types, with dilated cardiomyopathy (DCM) being the most common. Dysregulation of alternative splicing is one of the major underlying causes of cardiomyopathies.

- The impact of isoform switching remains poorly understood.
- Using long-read nanopore sequencing, we mapped full-length transcriptome of left ventricular tissue from DCM and ICM patients.
- Of the 11 prototypical sarcomere genes examined, 10 displayed significant shifts in isoform expression, including the tropomyosins. TPM1 and TPM3 showed a marked increase in heart failure.
- These results highlight a coordinated shift in the expression of heart-specific gene isoforms in the context of heart failure."

Please note that we have changed the article type to Resource. We note that you agree with the publication of the RPF.

I look forward to receiving your revised manuscript.

Yours sincerely,

Lise Roth

Lise Roth, PhD

Senior Editor

EMBO Molecular Medicine

The authors addressed the remaining editorial issues.

12th Dec 2025

Dear Prof. Meder,

Thank you for submitting your revised files. I am pleased to inform you that your manuscript is accepted for publication and is now being sent to our publisher to be included in the next available issue of EMBO Molecular Medicine.

You may qualify for financial assistance for your publication charges - either via a Springer Nature fully open access agreement or an EMBO initiative. Check your eligibility: <https://link.springer.com/journal/44321/how-to-publish-with-us>

With kind regards,

Lise Roth

>>> Please note that it is EMBO Molecular Medicine policy for the transcript of the editorial process (containing referee reports and your response letter) to be published as an online supplement to each paper. If you do NOT want this, you will need to inform the Editorial Office via email immediately. More information is available here: <https://link.springer.com/partners/embo-press/editorial-policies#Peer%20review>